# Accelerating Diffusion Large Language Models with SlowFast Sampling: The Three Golden Principles

**Qingyan Wei**[1]    **Yaojie Zhang**[1]    **Zhiyuan Liu**[1]    **Puyu Zeng**[1]
**Yuxuan Wang**[1]    **Biqing Qi**[2]    **Dongrui Liu**[2]    **Linfeng Zhang**[1†]

[1]EPIC Lab, Shanghai Jiao Tong University    [2]Shanghai Artificial Intelligence Laboratory

**Code:** `https://github.com/LiangrunFlora/Slow-Fast-Sampling`

## Abstract

Diffusion-based language models (dLLMs) have emerged as a promising alternative to traditional autoregressive LLMs by enabling parallel token generation and significantly reducing inference latency. However, existing sampling strategies for dLLMs, such as confidence-based or semi-autoregressive decoding, often suffer from static behavior, leading to suboptimal efficiency and limited flexibility. In this paper, we propose **SlowFast Sampling**, a novel dynamic sampling strategy that adaptively alternates between exploratory and accelerated decoding stages. Our method is guided by three golden principles: *certainty principle*, *convergence principle*, and *positional principle*, which govern when and where tokens can be confidently and efficiently decoded. We further integrate our strategy with dLLM-Cache to reduce redundant computation. Extensive experiments across various benchmarks demonstrate the efficiency of our method. Specifically, on the GPQA benchmark, SlowFast Sampling achieves up to $15.63\times$ speedup on LLaDA with minimal accuracy drop, and up to $34.22\times$ when combined with caching. Notably, our approach outperforms strong autoregressive baselines like LLaMA3 8B in throughput, demonstrating that well-designed sampling can unlock the full potential of dLLMs for fast and high-quality generation.

## 1 Introduction

Large Language Models (LLMs) (Zhao et al., 2025) have rapidly become cornerstone technologies in artificial intelligence, demonstrating remarkable capabilities across a diverse range of natural language understanding and generation tasks. However, the prevalent autoregressive nature of most LLMs, where tokens are generated sequentially one after another, introduces significant inference latency, particularly for long sequences. To address this inherent bottleneck, diffusion-based LLMs (dLLMs) (Ye et al., 2025; Nie et al., 2025b) have emerged as a promising alternative paradigm. These models are capable of generating multiple tokens in parallel, departing from the strict token-by-token process. This parallel decoding capability offers the distinct advantage of potentially accelerating text generation significantly, positioning dLLMs as a compelling and forward-looking direction for efficient language model inference.

However, current ways of sampling with dLLMs often don't perform as well as they could. Common methods include confidence-based selection (Chang et al., 2022) like Fast-dLLM (Wu et al., 2025), where tokens exceeding a confidence threshold are selected for decoding. Another popular method, semi-autoregressive decoding (Arriola et al., 2025), divides the sequence into fixed blocks and decodes within them. Unfortunately, these methods frequently yield unsatisfactory results (*i.e.*, significant accuracy drop when decoding many tokens in parallel), and are characterized by a static, constant sampling speed throughout the generation process. This lack of flexibility highlights the need for a more dynamic sampling approach: one that can smartly decide how many tokens to sample at each step and where these tokens should be located in the sequence.

---

[†] indicates the corresponding author.

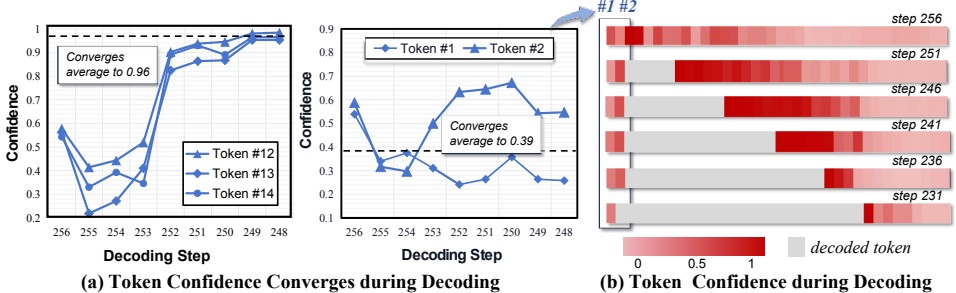

Figure 1: **The Three Golden Principles for sampling in diffusion LLMs.** (a) *Convergence Principle*: As decoding proceeds, the confidence values of tokens largely converge to high values, while a few tokens converge to lower values. (b) The confidence map over 256 diffusion steps: High-confidence tokens (in deep red) emerge progressively and are preferentially decoded (*the Certainty Principle*), while selection tends to cluster in contiguous regions (*the Positional Principle*), enabling cache reuse and acceleration.

Motivated by these limitations, we introduce a novel dynamic sampling approach designed to accelerate dLLMs, aiming to unlock the real potential of dLLMs under high-parallel decoding. As illustrated in Figure 1, our method is guided by three core observations, which we formulate as the **Three Golden Principles** for effective acceleration:

- **The Certainty Principle:** Tokens exhibiting higher confidence are inherently more determined. Consequently, they are more likely to be decoded correctly early in the process and require less adjustment in subsequent diffusion steps.
- **The Convergence Principle:** As the diffusion process unfolds and tokens are progressively refined, the semantic meaning of many tokens stabilizes, and their associated confidence scores converge towards a steady value. This convergence indicates that these tokens have largely settled into their final form and require minimal further refinement.
- **The Positional Principle:** We observe that even without explicit constraints, the model's sampling preferences often gravitate towards tokens in specific, frequently neighboring or clustered, positions. This inherent positional bias can be strategically exploited. For instance, parts of the sequence can be effectively cached, leading to significant acceleration gains.

Integrating these principles, we propose SlowFast Sampling with two phases: an Exploratory Stage and an Accelerated Decoding Stage. In the exploratory stage, the model loosely decodes to locate spans with emerging certainty and convergence. The accelerated stage then parallelly decodes these high-certainty tokens, reducing effort on already determined parts. This division yields significant speedups, reaching $15.63\times$ on LLaDA and up to $34.22\times$ when combined with **dLLM-Cache** (Liu et al., 2025) on the GPQA benchmark, with minimal accuracy loss. Our contributions are threefold:

1. We propose three golden principles based on token certainty, convergence, and positional influence, which critically govern effective and efficient sampling in dLLMs.

2. Building on these principles, we introduce SlowFast Sampling, a novel two-stage dynamic strategy specifically designed to leverage these principles for optimal acceleration of dLLM.

3. Through experiments on various benchmarks, we demonstrate that SlowFast Sampling achieves significant inference acceleration (*e.g.*, up to **$15.63\times$** on LLaDA with SlowFast Sampling alone, and up to **$34.22\times$** when combined with dLLM-Cache on the GPQA dataset) without compromising response quality, thereby offering a superior speed-quality trade-off compared to baseline and simpler sampling methods.

## 2 RELATED WORK

### 2.1 DIFFUSION MODELS FOR LANGUAGE

Diffusion Models (DMs) (Sohl-Dickstein et al., 2015; Ho et al., 2020; Song et al., 2021) have revolutionized generative modeling, particularly in continuous domains like images (Rombach et al.,

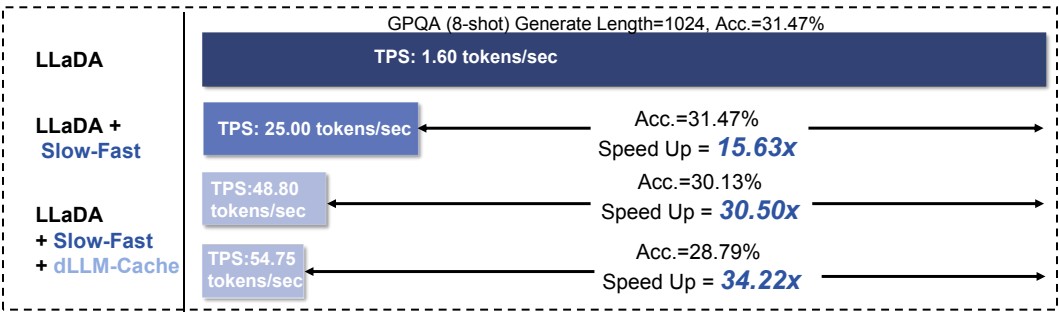

Figure 2: **Throughput and accuracy comparison on GPQA (8-shot, Length=1024) on LLaDA with our method,** including (1) vanilla decoding, (2) SlowFast Sampling, and (3) SlowFast Sampling further enhanced by dLLM-Cache. Compared to the vanilla setting, SlowFast Sampling alone achieves a **15.63×** speedup while maintaining comparable accuracy. With dLLM-Cache, throughput improves further to **54.75 tokens/sec** (up to **34.22×** speedup), with only minor drops in accuracy.

2022; Peebles & Xie, 2023; Ma et al., 2025b). However, adapting these models to discrete data such as text presents unique challenges due to its discrete nature. A promising approach in discrete diffusion models involves Masked Diffusion Models (MDMs) (Austin et al., 2021; Lou et al., 2023; Shi et al., 2024; Nie et al., 2025a;b; Hoogeboom et al., 2021; Campbell et al., 2022), which iteratively predict masked tokens based on their context. These advancements have transformed text generation, offering a compelling alternative to autoregressive paradigms in LLMs. Notable examples include LLaDA (Nie et al., 2025b), an 8B MDM trained from scratch with a bidirectional Transformer, and Dream (Ye et al., 2025), which initializes from pre-trained ARM weights. Both models demonstrate performance comparable to similarly-sized ARMs like LLaMA3 8B (Dubey et al., 2024). Their bidirectional architecture may overcome ARM limitations such as the reversal curse (Berglund et al., 2023), making diffusion a competitive alternative for foundational LLMs.

## 2.2 ACCELERATION METHODS FOR DIFFUSION-BASED LLMS

The high inference latency of dLLMs, primarily due to their iterative denoising process (Nie et al., 2025b; Ye et al., 2025), has spurred research into acceleration techniques. Various strategies have been developed, mainly including caching mechanisms and advanced sampling techniques.

**Caching Mechanisms.** Feature caching reduces redundant computations by reusing intermediate features. dLLM-Cache (Liu et al., 2025) combines long-interval prompt and short-interval response caching with a V-verify mechanism for faster inference. Sparse-dLLM (Song et al., 2025) applies dynamic cache eviction with sparse attention, retaining only salient tokens to cut memory and boost speed. dKV-Cache (Ma et al., 2025a) adopts delayed KV caching to reuse decoded tokens' representations, achieving 2–10× acceleration.

**Advanced Sampling Techniques.** Optimizing the sampling process itself is another major direction for accelerating dLLMs. Low-confidence remasking (Chang et al., 2022; Nie et al., 2025b) prioritizes high-confidence tokens to speed up convergence; semi-autoregressive (Nie et al., 2025b; Arriola et al., 2025) remasking divides sequences into blocks, applying random and low-confidence strategies. Additionally, GtR (Yan et al., 2026) accelerates MAR models through a hierarchical structure-to-detail sampling strategy, while exact simulation methods for MDMs like the first-hitting sampler (Zheng et al., 2024) have made progress in reducing sampling steps or enhancing per-step efficiency.

## 3 METHODOLOGY

### 3.1 PRELIMINARY

**Inference Process of Diffusion Large Language Models.** Diffusion Large Language Models (dLLMs) generate text $\mathbf{y} = (y_1, \ldots, y_L)$ from a prompt $\mathbf{c} = (c_1, \ldots, c_M)$ through an iterative denoising process. The model refines an intermediate state $\mathbf{y}^{(k)} \in \mathcal{T}^L$ over $N$ discrete steps, from $k = N$ to $k = 0$, where $\mathcal{T}$ is the token vocabulary. The process begins with a fully masked sequence:

$$\mathbf{y}^{(N)} = \big(\underbrace{[\text{MASK}], \ldots, [\text{MASK}]}_{L \text{ times}}\big) \tag{1}$$

where [MASK] $\in \mathcal{T}$ is the special mask token.

At each step $k \in \{N, N-1, \ldots, 1\}$, a mask predictor $p_\theta$ estimates the original clean sequence $\mathbf{r}_0 = (r_{0,1}, \ldots, r_{0,L})$ from the noisy state $\mathbf{y}^{(k)}$ and prompt $\mathbf{c}$:

$$P_\theta(\mathbf{r}_0|\mathbf{c}, \mathbf{y}^{(k)}) \tag{2}$$

An estimate of the clean sequence at step $k$, $\hat{\mathbf{r}}_0^{(k)}$, is obtained through greedy decoding:

$$\hat{r}_{0,i}^{(k)} = \arg\max_{v \in \mathcal{T}} P_\theta(r_{0,i} = v|\mathbf{c}, \mathbf{y}^{(k)}) \quad \forall i \in \{1, \ldots, L\} \tag{3}$$

Although the predictor $p_\theta$ can decode all masked tokens [MASK] in one step, to ensure high-quality generation, dLLM adopts a multi-step decoding process. At each step, the remask strategy refines the tokens. The transition to the next state $\mathbf{y}^{(k-1)}$ is governed by a sampling strategy $S$:

$$\mathbf{y}^{(k-1)} = S(\hat{\mathbf{r}}_0^{(k)}, \mathbf{y}^{(k)}, k) \tag{4}$$

This iterative denoising process is a sampling procedure. Our work aims to improve the sampling efficiency of dLLM inference, which can be computationally expensive due to the $N$ sequential steps. Using LLaDA (Nie et al., 2025b) as an example, we describe its core sampling strategies. In LLaDA, time steps are defined as $t_k = k/N$ and $t_{k-1} = (k-1)/N$.

LLaDA explores various strategies for the transition function $S$ in Equation 4, which differ mainly in how tokens from $\hat{\mathbf{r}}_0^{(k)}$ update $\mathbf{y}^{(k)}$ to form $\mathbf{y}^{(k-1)}$, especially for positions that were [MASK] in $\mathbf{y}^{(k)}$:

**Random Remasking.** In this strategy, for each position $i$:
If $y_i^{(k)} \neq$ [MASK], then $y_i^{(k-1)} = y_i^{(k)}$ (known tokens are preserved).
If $y_i^{(k)} =$ [MASK], then $y_i^{(k-1)}$ is set to $\hat{r}_{0,i}^{(k)}$ with probability $1 - \frac{k-1}{k}$, and remains [MASK] with probability $\frac{k-1}{k}$, ensuring the expected number of masked tokens aligns with the noise schedule.

**Low-Confidence Remasking.** This deterministic strategy aims to improve sample quality by selectively unmasking tokens. For each position $i$, if $y_i^{(k)} =$ [MASK], the model predicts $\hat{r}_{0,i}^{(k)}$ and computes its confidence. Specifically, the confidence $c_i$ is given by:

$$c_i = P_\theta(\hat{r}_{0,i}^{(k)}|\mathbf{c}, \mathbf{y}^{(k)}). \tag{5}$$

If $y_i^{(k)} \neq$ [MASK], then $c_i = 1$.

The target number of unmasked tokens for state $\mathbf{y}^{(k-1)}$ is given by:

$$n_{un} = \lfloor L(1 - t_{k-1}) \rfloor = \lfloor L\left(1 - \frac{k-1}{N}\right) \rfloor. \tag{6}$$

The tokens corresponding to the $n_{un}$ highest confidences are unmasked in $\mathbf{y}^{(k-1)}$, while the remaining positions are set to [MASK].

**Semi-Autoregressive Remasking.** This strategy is used in LLaDA after Supervised Fine-Tuning. The sequence is divided into blocks, and generation proceeds block by block from left to right. Within each block, the random remasking or the low-confidence remasking strategy is applied iteratively to denoise the block before moving to the next.

The choice of sampling strategy $S$, along with the number of steps $N$, significantly affects both the generation quality and latency.

### 3.2 THREE GOLDEN PRINCIPLES OF dLLMS

Building upon the iterative denoising process outlined in Section 3.1, our empirical analysis of dLLM behavior, particularly with models like LLaDA, has revealed consistent patterns in token generation. These patterns, which we term the **Three Golden Principles**, form the bedrock of our proposed acceleration strategy. They describe how token certainty, convergence, and positional effects interact during the diffusion process, offering key insights into optimizing sampling.

**The Certainty Principle: High Confidence Indicates Determination.** We observe that tokens predicted with high confidence by the mask predictor $p_\theta$ are significantly more likely to be part of the

final, correct sequence and tend to remain unchanged in subsequent denoising steps. The confidence for a predicted token $\hat{r}_{0,i}^{(k)}$ at position $i$ and step $k$ is given by $P_\theta(\hat{r}_{0,i}^{(k)}|\mathbf{c}, \mathbf{y}^{(k)})$. As illustrated in Figure 1 (b), at any given step $k$, a subset of tokens typically exhibits substantially higher confidence scores compared to others. These high-confidence tokens are prime candidates for early "acceptance" or less frequent resampling.

***Implication for Acceleration:*** By prioritizing tokens that quickly reach a high confidence threshold, we can reduce redundant computations on already determined parts of the sequence.

**The Convergence Principle: Tokens Stabilize Over Iterations.** During the iterative refinement process, individual tokens (both their predicted identity $\hat{r}_{0,i}^{(k)}$ and their confidence $P_\theta(\hat{r}_{0,i}^{(k)}|\mathbf{c}, \mathbf{y}^{(k)})$) undergo a period of fluctuation before settling. As shown in Figure 1 (a), a representative token might initially change its predicted identity and confidence across several early diffusion steps. However, as $k$ decreases, the token's confidence converges are often to a stable value. This convergence signals that the model has formed a consistent belief about the token's identity within its current context.

***Implication for Acceleration:*** Tokens that have demonstrated convergence (*i.e.*, stable identity and confidence over a window of recent steps) are less likely to change. Aggressively decoding can prevent unnecessary re-evaluation, thereby speeding up the process.

**The Positional Principle: Decoding Exhibits Regional Preferences.** Beyond individual token behaviors, we find that the generation process often exhibits spatial patterns. High-confidence and early-converging tokens do not appear randomly scattered throughout the sequence. Instead, they frequently emerge in contiguous blocks or localized regions, as suggested by Figure 1 (b). This phenomenon might be due to local semantic dependencies or the influence of strongly contextualized parts of the prompt $\mathbf{c}$. For example, after a few initial steps, a particular span of tokens might collectively achieve high confidence and stability, while other regions remain largely masked or uncertain. The model appears to focus its decoding efforts on specific segments at different stages.

***Implication for Acceleration:*** Recognizing these regions allows for targeted decoding. Instead of uniformly processing all tokens, computational resources can be concentrated on regions that are currently most amenable to decoding.

These three principles collectively indicate that a one-size-fits-all, static sampling strategy is inherently inefficient. They motivate a dynamic approach where the number of tokens sampled, their selection criteria, and their locations are adapted based on the evolving state of the generated sequence. Our SlowFast Sampling methodology, detailed in Section 3.3, is designed to explicitly leverage these observations to achieve significant inference speedups.

### 3.3 SLOWFAST SAMPLING

**1. Exploratory Stage (Slow Phase): Identifying the Next Stable Region.** As illustrated in Figure 3, the primary goal of this stage is to cautiously advance decoding while identifying a promising, stable region for subsequent rapid processing. Starting from $s_{cycle}$ and extending to the end of the full sequence $L$, this stage operates as follows for a limited number of dLLM steps:

- **Cautious Decoding:** At each dLLM step $k$ within this stage, we perform a conservative decoding update. We select the top-$k_{slow}$ tokens within the current exploratory window $[s_{cycle}, L]$ that exhibit the highest confidence $P_\theta(\hat{r}_{0,i}^{(k)}|\mathbf{c}, \mathbf{y}^{(k)})$, and these are unmasked to form part of $\hat{\mathbf{r}}_0^{(k)}$ (as per Equation 3 for these tokens, followed by Equation 4).

- **End Point of Convergence Prediction:** Concurrently, the model predicts a end point of candidate convergence, $e_{cand}^{(k)}$. This is defined as the furthest token position $i \in [s_{cycle}, L]$ for which the predicted confidence $P_\theta(\hat{r}_{0,i}^{(k)}|\mathbf{c}, \mathbf{y}^{(k)})$ exceeds a minimum confidence threshold $\tau_{min\_conf}$. Mathematically:

$$e_{cand}^{(k)} = \max\{i \mid i \in [s_{cycle}, L] \wedge P_\theta(\hat{r}_{0,i}^{(k)}|\mathbf{c}, \mathbf{y}^{(k)}) > \tau_{min\_conf}\} \qquad (7)$$

This $e_{cand}^{(k)}$ represents an estimate of how far into the sequence the model can confidently decode at the current step $k$.

- **Stability Check:** Throughout the Exploratory Stage, candidate convergence horizons $e_{cand}^{(j)}$ predicted at each dLLM step $j$ are tracked. Let $H_W(k_s) = \{e_{cand}^{(l)} \mid l \in [\max(1, k_s - W_{hist}+1), k_s]\}$

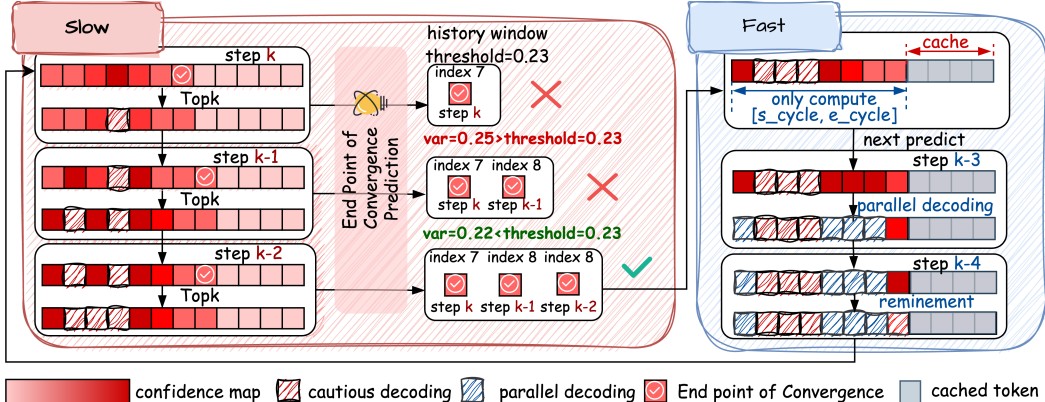

Figure 3: **Overview of the SlowFast Sampling Pipeline.** The method alternates between a **Slow (Exploratory) stage** and a **Fast (Accelerated) stage** for efficient token generation. In the Slow phase (left), the model conducts *cautious decoding* by selecting top-$k$ high-confidence tokens per step while continuously predicting the *End Point of Convergence* and calculating confidence variance across a history window. Once variance drops below threshold (*e.g.*, $0.22 < 0.23$), the corresponding region $[s_{cycle}, e_{cycle}]$ is considered stable. In the Fast phase (right), this stable span is decoded in parallel with aggressive unmasking of high-confidence tokens, while tokens beyond the span are temporarily skipped and their results *cached* for reuse. This alternating structure reduces redundant computation and accelerates decoding while maintaining output quality.

be the set of candidate horizons in the sliding window of the latest $W_{hist}$ predictions ending at exploratory step $k_s$. Stability is achieved, concluding this stage, when $\mathrm{Var}(H_W(k_s)) < \sigma^2_{stable}$ for a window where $k_s \geq W_{hist}$. The exploratory stage ends at step $k_{final}$, defined as:

$$k_{final} = \min\left(\{k_s \mid W_{hist} \leq k_s \leq K_{max} \wedge \mathrm{Var}(H_W(k_s)) < \sigma^2_{stable}\} \cup \{K_{max}\}\right) \quad (8)$$

If the stability criterion is not met by $K_{max}$ (the maximum allotted exploratory steps), then $k_{final} = K_{max}$. The cycle's convergence horizon $e_{cycle}$, is then set to the mean of the candidate horizons in the final window $H_W(k_{final})$:

$$e_{cycle} = \mathrm{Mean}(H_W(k_{final})) = \frac{1}{|H_W(k_{final})|} \sum_{e \in H_W(k_{final})} e \quad (9)$$

Once this stability criterion is met, the Exploratory Stage concludes. The endpoint for the subsequent Accelerated Decoding Stage, $e_{cycle}$, is set to the last recorded candidate horizon, $e^{(k)}_{cand}$. If stability is not achieved within a maximum number of exploratory steps, $e_{cycle}$ can be set to a conservatively determined position or the process might default to a full-sequence cautious decode for that cycle.

**2. Accelerated Decoding Stage (Fast Phase): Rapid Parallel Refinement.** As shown in the right half of Figure 3, once a stable region $[s_{cycle}, e_{cycle}]$ is identified, this stage aims to rapidly denoise tokens within this span, while efficiently handling tokens outside it:

- **Out-of-Span Caching:** For tokens outside the identified span (*i.e.*, positions $i > e_{cycle}$), if their current predicted confidence is low (*e.g.*, below $\tau_{min\_conf}$), their predicted values $\hat{r}^{(k)}_{0,i}$ from one dLLM step within this stage are computed and then cached. These cached values can be reused in subsequent dLLM steps for these positions, provided they remain outside an active decoding span, thus saving redundant computations.

- **In-Span Parallel Decoding:** Within the span $[s_{cycle}, e_{cycle}]$, an aggressive parallel decoding is attempted. All tokens $y^{(k)}_i = $ [MASK] for $i \in [s_{cycle}, e_{cycle}]$ for which the predicted confidence $P_\theta(\hat{r}^{(k)}_{0,i}|\mathbf{c}, \mathbf{y}^{(k)})$ exceeds the high certainty threshold $\tau_{high\_conf}$, *i.e.*,

$$P_\theta(\hat{r}^{(k)}_{0,i}|\mathbf{c}, \mathbf{y}^{(k)}) > \tau_{high\_conf} \quad (10)$$

are unmasked by setting their value to the corresponding prediction $\hat{r}^{(k)}_{0,i}$. This update is performed simultaneously for all such qualifying tokens in a single conceptual step to form $\hat{\mathbf{r}}^{(k)}_0$.

Table 1: **Performance of LLaDA 8B and Dream 7B with SlowFast Sampling** on 8 benchmarks.

| Task | Method | Inference Efficiency | | Performance | Method | Inference Efficiency | | Performance |
|---|---|---|---|---|---|---|---|---|
| | | TPS↑ | Speed↑ | Score↑ | | TPS↑ | Speed↑ | Score↑ |
| **Mathematics & Science** | | | | | | | | |
| GSM8K | LLaDA | 4.55 | 1.00× | 69.83 | Dream | 8.16 | 1.00× | 77.02 |
| | + Fast-dLLM (Parallel) | $7.45_{+2.90}$ | $1.64\times_{+0.64}$ | $69.60_{-0.23}$ | + Fast-dLLM (Parallel) | $12.72_{+4.56}$ | $1.55\times_{+0.55}$ | $73.09_{-3.93}$ |
| | + SlowFast | $14.57_{+10.02}$ | $3.20\times_{+2.20}$ | $69.59_{-0.27}$ | + SlowFast | $17.15_{+8.99}$ | $2.10\times_{+1.10}$ | $76.50_{-0.52}$ |
| GPQA | LLaDA | 3.31 | 1.00× | 31.47 | Dream | 5.43 | 1.00× | 35.93 |
| | + Fast-dLLM (Parallel) | $11.72_{+8.41}$ | $3.54\times_{+2.54}$ | $32.13_{+0.66}$ | + Fast-dLLM (Parallel) | $15.88_{+10.45}$ | $2.92\times_{+1.92}$ | $31.01_{-4.92}$ |
| | + SlowFast | $16.36_{+13.05}$ | $4.94\times_{+3.94}$ | $31.91_{+0.44}$ | + SlowFast | $16.56_{+11.13}$ | $3.05\times_{+2.05}$ | $35.94_{+0.01}$ |
| Math | LLaDA | 5.14 | 1.00× | 30.16 | Dream | 8.48 | 1.00× | 38.68 |
| | + Fast-dLLM (Parallel) | $8.94_{+3.80}$ | $1.74\times_{+0.74}$ | $30.52_{+0.36}$ | + Fast-dLLM (Parallel) | $22.51_{+14.03}$ | $2.65\times_{+1.65}$ | $34.14_{-4.54}$ |
| | + SlowFast | $11.27_{+6.13}$ | $2.19\times_{+1.19}$ | $29.64_{-0.52}$ | + SlowFast | $23.00_{+14.52}$ | $2.71\times_{+1.71}$ | $38.24_{-0.44}$ |
| **General Tasks** | | | | | | | | |
| MMLU-pro | LLaDA | 9.16 | 1.00× | 23.30 | Dream | 14.97 | 1.00× | 24.14 |
| | + Fast-dLLM (Parallel) | $15.62_{+6.46}$ | $1.71\times_{+0.71}$ | $23.50_{+0.20}$ | + Fast-dLLM (Parallel) | $25.50_{+10.53}$ | $1.70\times_{+0.70}$ | $19.53_{-4.61}$ |
| | + SlowFast | $23.14_{+13.98}$ | $2.53\times_{+1.53}$ | $23.85_{+0.55}$ | + SlowFast | $22.80_{+7.83}$ | $1.52\times_{+0.52}$ | $22.91_{-1.23}$ |
| MMLU | LLaDA | 5.02 | 1.00× | 62.11 | Dream | 8.46 | 1.00× | 72.61 |
| | + Fast-dLLM (Parallel) | $12.94_{+7.92}$ | $2.58\times_{+1.58}$ | $61.67_{-0.44}$ | + Fast-dLLM (Parallel) | $17.39_{+8.93}$ | $2.05\times_{+1.05}$ | $64.59_{-8.02}$ |
| | + SlowFast | $16.81_{+11.79}$ | $3.35\times_{+2.35}$ | $66.56_{+4.45}$ | + SlowFast | $18.43_{+9.97}$ | $2.18\times_{+1.18}$ | $75.13_{+2.52}$ |
| BBH | LLaDA | 4.04 | 1.00× | 44.97 | Dream | 6.93 | 1.00× | 51.83 |
| | + Fast-dLLM (Parallel) | $10.73_{+6.69}$ | $2.66\times_{+1.66}$ | $44.13_{-0.84}$ | + Fast-dLLM (Parallel) | $27.84_{+20.91}$ | $4.01\times_{+3.01}$ | $52.28_{+0.45}$ |
| | + SlowFast | $21.19_{+17.15}$ | $5.24\times_{+4.24}$ | $44.60_{-0.37}$ | + SlowFast | $28.14_{+21.21}$ | $4.06\times_{+3.06}$ | $50.55_{-1.28}$ |
| **Code** | | | | | | | | |
| MBPP | LLaDA | 4.98 | 1.00× | 40.80 | Dream | 8.92 | 1.00× | 54.20 |
| | + Fast-dLLM (Parallel) | $8.15_{+3.17}$ | $1.64\times_{+0.64}$ | $40.80_{+0.00}$ | + Fast-dLLM (Parallel) | $22.87_{+13.95}$ | $2.56\times_{+1.56}$ | $49.40_{-4.80}$ |
| | + SlowFast | $13.32_{+8.34}$ | $2.67\times_{+1.67}$ | $41.00_{+0.20}$ | + SlowFast | $29.07_{+20.15}$ | $3.26\times_{+2.26}$ | $54.60_{+0.40}$ |
| HumanEval | LLaDA | 11.24 | 1.00× | 31.71 | Dream | 11.49 | 1.00× | 34.15 |
| | + Fast-dLLM (Parallel) | $23.05_{+11.81}$ | $2.05\times_{+1.05}$ | $32.32_{+0.61}$ | + Fast-dLLM (Parallel) | $24.33_{+12.84}$ | $2.12\times_{+1.12}$ | $32.92_{-1.23}$ |
| | + SlowFast | $35.46_{+24.22}$ | $3.15\times_{+2.15}$ | $33.54_{+1.83}$ | + SlowFast | $25.38_{+13.89}$ | $2.21\times_{+1.21}$ | $35.36_{+1.21}$ |

- **Fallback Top-k Refinement:** If the number of tokens meeting the $\tau_{high\_conf}$ criterion within the span is insufficient to make substantial progress (*e.g.*, less than one), we revert to a more conservative update for this dLLM step within the span. Specifically, we select the top-$k_{fast}$ tokens within $[s_{cycle}, e_{cycle}]$ based on confidence and unmask them. This ensures steady progress even when widespread high certainty is not yet achieved.

After the Accelerated Decoding Stage completes, the starting position for the next cycle's Exploratory Stage is updated to $s_{cycle} \leftarrow e_{cycle}$. This cyclical process of exploration and accelerated decoding continues until the entire sequence is generated. This SlowFast approach dynamically adapts the decoding focus and intensity, leveraging the Certainty and Positional principles to identify promising regions and the Convergence principle to stabilize them efficiently.

## 4 EXPERIMENT

### 4.1 EXPERIMENT SETTINGS

**Implementation Details** To evaluate the effectiveness of our proposed dynamic sampling approach SlowFast Sampling, we conducted experiments on representative dLLMs: LLaDA 8B (Nie et al., 2025b) and Dream 7B (Ye et al., 2025), focusing on measuring the inference acceleration across various benchmarks. All experiments were conducted on NVIDIA RTX 4090 GPUs.

**Evaluation Metrics** We evaluated sampling acceleration and generation quality of SlowFast Sampling using quantitative metrics. Inference speed is measured in Tokens Per Second (TPS), indicating the average number of tokens generated per second. Generation quality is assessed using task-specific metrics, *e.g.*, accuracy on GSM8K, reflecting the model's performance under inference acceleration.

### 4.2 MAIN RESULTS

**Performance and efficiency gains across models.** Table 1 reports throughput and model performance for both LLaDA 8B and Dream 7B, with and without SlowFast Sampling. These results demonstrate that our method brings significant improvements in inference efficiency and achieves lossless acceleration in most cases. In our experiments, the key hyperparameters of SlowFast Sampling, $\tau_{min\_conf}$ and $\tau_{high\_conf}$, were set to $0.1$ and $0.85$, respectively. The remaining hyperparameters in our method were set as follows: maximum exploratory steps $K_{max} = 8$, sliding window size

Table 2: **Performance of LLaDA 8B and Dream 7B with SlowFast Sampling and dLLM-Cache.**

| Task | Method | Inference Efficiency | | Performance | Method | Inference Efficiency | | Performance |
|---|---|---|---|---|---|---|---|---|
| | | TPS↑ | Speed(TPS)↑ | Score↑ | | TPS↑ | Speed(TPS)↑ | Score↑ |
| **Mathematics & Science** | | | | | | | | |
| GSM8K | LLaDA | 4.55 | 1.00× | 69.83 | Dream | 8.16 | 1.00× | 77.02 |
| | + Fast-dLLM (Parallel+Cache) | $15.50_{+10.95}$ | $3.41\times_{+2.41}$ | $68.77_{-1.06}$ | + Fast-dLLM (Parallel+Cache) | $31.07_{+22.91}$ | $3.81\times_{+2.81}$ | $69.45_{-7.57}$ |
| | + SlowFast + Cache | $26.99_{+22.44}$ | $5.93\times_{+4.93}$ | $69.60_{-0.23}$ | + SlowFast + Cache | $46.17_{+38.01}$ | $5.66\times_{+4.66}$ | $72.10_{-4.92}$ |
| GPQA | LLaDA | 3.31 | 1.00× | 31.47 | Dream | 5.43 | 1.00× | 35.93 |
| | + Fast-dLLM (Parallel+Cache) | $30.21_{+26.90}$ | $9.13\times_{+8.13}$ | $33.03_{+1.56}$ | + Fast-dLLM (Parallel+Cache) | $40.98_{+35.55}$ | $7.54\times_{+6.54}$ | $32.37_{-3.56}$ |
| | + SlowFast + Cache | $29.06_{+25.75}$ | $8.78\times_{+7.78}$ | $33.48_{+2.01}$ | + SlowFast + Cache | $39.63_{+34.20}$ | $7.30\times_{+6.30}$ | $34.82_{-1.11}$ |
| Math | LLaDA | 5.14 | 1.00× | 30.16 | Dream | 8.48 | 1.00× | 38.68 |
| | + Fast-dLLM (Parallel+Cache) | $21.50_{+16.36}$ | $4.18\times_{+3.18}$ | $28.34_{-1.82}$ | + Fast-dLLM (Parallel+Cache) | $46.80_{+38.32}$ | $5.52\times_{+4.52}$ | $33.24_{-5.44}$ |
| | + SlowFast + Cache | $26.50_{+21.36}$ | $5.16\times_{+4.16}$ | $29.42_{-0.74}$ | + SlowFast + Cache | $56.44_{+47.96}$ | $6.66\times_{+5.66}$ | $37.10_{-1.58}$ |
| **General Tasks** | | | | | | | | |
| MMLU-pro | LLaDA | 9.16 | 1.00× | 23.30 | Dream | 14.97 | 1.00× | 24.14 |
| | + Fast-dLLM (Parallel+Cache) | $27.85_{+18.69}$ | $3.04\times_{+2.04}$ | $27.66_{+4.36}$ | + Fast-dLLM (Parallel+Cache) | $42.73_{+27.76}$ | $2.85\times_{+1.85}$ | $22.57_{-1.57}$ |
| | + SlowFast + Cache | $33.38_{+24.22}$ | $3.64\times_{+2.64}$ | $25.53_{+2.23}$ | + SlowFast + Cache | $43.50_{+28.53}$ | $2.91\times_{+1.91}$ | $21.81_{-2.33}$ |
| MMLU | LLaDA | 5.02 | 1.00× | 62.11 | Dream | 8.46 | 1.00× | 72.61 |
| | + Fast-dLLM (Parallel+Cache) | $32.36_{+27.34}$ | $6.45\times_{+5.45}$ | $61.45_{-0.66}$ | + Fast-dLLM (Parallel+Cache) | $44.73_{+36.27}$ | $5.28\times_{+4.28}$ | $64.46_{-8.15}$ |
| | + SlowFast + Cache | $38.42_{+33.40}$ | $7.65\times_{+6.65}$ | $61.20_{-0.91}$ | + SlowFast + Cache | $44.18_{+35.72}$ | $5.22\times_{+4.22}$ | $71.57_{-1.04}$ |
| BBH | LLaDA | 4.04 | 1.00× | 44.97 | Dream | 6.93 | 1.00× | 51.83 |
| | + Fast-dLLM (Parallel+Cache) | $22.56_{+18.52}$ | $5.58\times_{+4.58}$ | $45.35_{+0.38}$ | + Fast-dLLM (Parallel+Cache) | $57.02_{+50.09}$ | $8.22\times_{+7.22}$ | $44.95_{-6.88}$ |
| | + SlowFast + Cache | $36.04_{+32.00}$ | $8.92\times_{+7.92}$ | $44.81_{-0.16}$ | + SlowFast + Cache | $70.20_{+63.27}$ | $10.13\times_{+9.13}$ | $48.24_{-3.59}$ |
| **Code** | | | | | | | | |
| MBPP | LLaDA | 4.98 | 1.00× | 40.80 | Dream | 8.92 | 1.00× | 54.20 |
| | + Fast-dLLM (Parallel+Cache) | $22.18_{+17.20}$ | $4.45\times_{+3.45}$ | $38.20_{-2.60}$ | + Fast-dLLM (Parallel+Cache) | $50.67_{+41.75}$ | $5.68\times_{+4.68}$ | $50.40_{-3.80}$ |
| | + SlowFast + Cache | $27.26_{+22.28}$ | $5.47\times_{+3.87}$ | $39.00_{-1.80}$ | + SlowFast + Cache | $69.48_{+60.56}$ | $7.79\times_{+6.79}$ | $51.00_{-3.20}$ |
| HumanEval | LLaDA | 11.24 | 1.00× | 31.71 | Dream | 11.49 | 1.00× | 34.15 |
| | + Fast-dLLM (Parallel+Cache) | $26.25_{+15.01}$ | $2.34\times_{+1.34}$ | $29.88_{-1.83}$ | + Fast-dLLM (Parallel+Cache) | $49.47_{+37.98}$ | $4.31\times_{+3.31}$ | $29.27_{-4.88}$ |
| | + SlowFast + Cache | $41.14_{+29.90}$ | $3.66\times_{+2.66}$ | $31.10_{-0.61}$ | + SlowFast + Cache | $47.86_{+36.37}$ | $4.17\times_{+3.17}$ | $35.36_{+1.21}$ |

| Sampling Strategy | TPS↑ | Score↑ |
|---|---|---|
| Autoregressive (AR) | 5.25 | 60.80 |
| Diffusion Sampling | 4.55 | 69.83 |
| Semi-Autoregressive | 5.44 | 66.41 |
| SlowFast Sampling | 9.87 | 69.59 |

| Method | TPS↑ | Speed(TPS)↑ | Accuracy↑ |
|---|---|---|---|
| LLaMA3 8B (Dubey et al., 2024) | 33.79 | 21.12× | 31.92 |
| LLaDA | $1.60_{-32.19}$ | 1.00× | $31.47_{-0.45}$ |
| + SlowFast | $25.00_{-8.79}$ | 15.63× | $31.47_{-0.45}$ |
| + SlowFast + Cache ($K_p = 100, K_r = 5$) | $48.80_{+15.01}$ | 30.50× | $30.13_{-1.79}$ |
| + SlowFast + Cache ($K_p = 500, K_r = 30$) | $54.75_{+20.96}$ | 34.22× | $28.79_{-3.13}$ |

Table 3: **Comparison of Sampling Strategies** on inference efficiency and generation performance.

Table 4: **Comparison of LLaDA 8B Base with other representative LLMs**. Compared to LLaMA3 8B, LLaDA with SlowFast Sampling and dLLM-Cache achieves significantly higher throughput (up to +20.96 TPS) while maintaining comparable accuracy.

$W_{hist} = 2$, and stable variance threshold $\sigma^2_{stable} = 1.0$, consistent with the default configuration used in Section 4.2 and ablation studies.

**Compatibility with dLLM-Cache.** Recently, several studies have explored leveraging feature cache to reduce the computational cost of dLLM inference. Our SlowFast Sampling is highly compatible with existing caching mechanisms and the integration can lead to higher acceleration compared to using either alone. Table 2 compares the performance and inference speed of LLaDA with and without applying our method in combination with dLLM-Cache. The results show that this integration can deliver higher throughput while maintaining comparable model performance.

**Comparison with Other Sampling Strategies.** We compared our method SlowFast Sampling with four alternative sampling strategies: diffusion sampling, diffusion semi-autoregressive sampling, autoregressive (AR) sampling and Fast-dLLM (Wu et al., 2025). Diffusion sampling adopts a remasking strategy to iteratively select token to decode in parallel, while AR sampling generates tokens strictly from left to right. The semi-autoregressive approach generates blocks left-to-right and applies the remasking strategy within each block. Fast-dLLM, in contrast, performs parallel decoding only based on token confidence, and we report results for both its parallel and cache-augmented variants. As shown in Table 1, 2 and 3, our method SlowFast Sampling achieves higher inference efficiency while maintaining competitive generation quality.

## 4.3 ABLATION STUDY

**Case Study of SlowFast Sampling in Sentence Generation.** As shown in Figure 4, this case study shows how SlowFast Sampling generates text by alternating slow and fast phases. In the slow phases, the model outputs a few reliable tokens such as subjects, verbs, or punctuation to anchor the sentence. The fast phases then produce longer spans of high-confidence tokens in one step, e.g., "*she has 9 - 4 = 5 yuan left*", ensuring both efficiency and correctness. This balance of cautious slow decoding

and confident fast decoding maintains accuracy while accelerating generation, making the approach intuitive and effective.

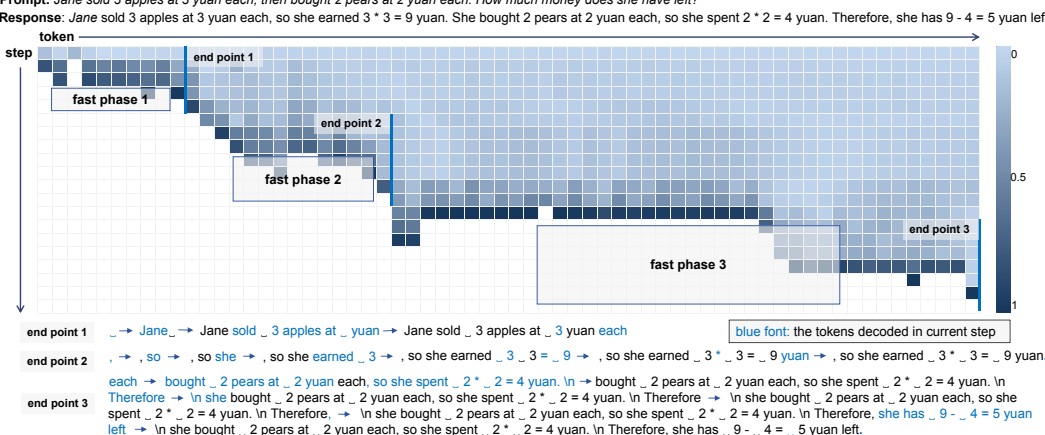

Figure 4: **Case study of sentence generation with SlowFast Sampling.** The figure illustrates the dynamic evolution of the confidence map across three phases, where exploratory and accelerated decoding are alternated. Blue tokens denote those generated in the current step, highlighting how the method gradually builds on high-confidence spans to produce a coherent final response.

**Effect of Hyperparameters in the Stability Check.** The stability check, which transitions the model from the slow to the fast phase, is governed by three hyperparameters whose effects are shown in Figure 5. The maximum number of exploratory steps, $K_{max}$, is for allowing the convergence horizon to stabilize; we find $K_{max} = 8$ provides sufficient exploration for high-quality generation without incurring excessive overhead. This is complemented by the sliding window size, $W_{hist}$. Since prediction changes and convergence occur rapidly, we find a smaller window of $W_{hist} = 2$ is effective, striking a strong balance between maintaining quality and maximizing inference speed. Finally, a strict stable variance threshold of $\sigma^2_{stable} = 1.0$ ensures that the accelerated phase is only triggered for genuinely stable regions, solidifying the reliability of our method.

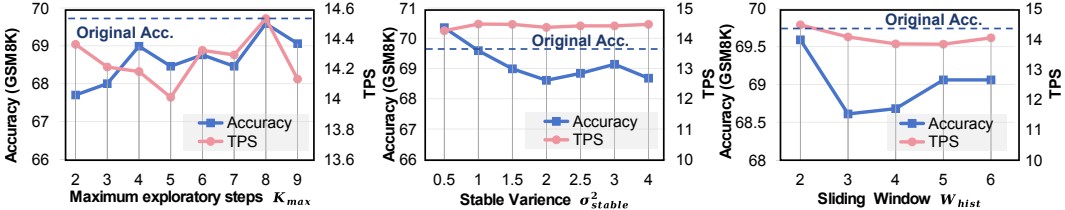

Figure 5: **The sensitivity study on hyper-parameters in the stability-Check.** Accuracy and TPS vary with $K_{max}$, $\sigma^2_{stable}$, and $W_{hist}$. The chosen defaults ($K_{max} = 8$, $\sigma^2_{stable} = 1.0$, $W_{hist} = 2$) offer strong speed-quality trade-offs.

**Outperforming Autoregressive LLMs in Inference Speed.** As shown in Table 4, when equipped with our SlowFast Sampling and dLLM-Cache, LLaDA Base not only accelerates significantly over its default setting but also **outperforms the autoregressive LLaMA3 8B** (Dubey et al., 2024) in inference speed (54.75 vs. 33.79 TPS), while maintaining comparable accuracy. This demonstrates that dLLMs, with proper sampling and optimization, can surpass traditional autoregressive models in both efficiency and practicality.

## 5 CONCLUSION

In this work, we present **SlowFast Sampling**, a dynamic and principled approach to accelerate diffusion-based large language models (dLLMs). By leveraging three key observations: token certainty, convergence, and positional bias, we design a two-stage decoding pipeline that adaptively balances exploration and efficient parallel decoding. Extensive experiments across benchmarks demonstrate that our method not only significantly improves inference speed (up to **15.63×** and up to **34.22×** combined with dLLM-Cache on the GPQA benchmark), but also maintains strong generation quality, outperforming even autoregressive LLMs in throughput. We believe this work marks an important step toward making dLLMs practical and competitive in real-world deployment.

## ACKNOWLEDGEMENT

This project is sponsored by CCF-Tencent Rhino-Bird Funds.

## REPRODUCIBILITY STATEMENT

We have made every effort to ensure the reproducibility of our results. The main text (Sections 3–4) provides detailed descriptions of the proposed SlowFast Sampling method, its integration with dLLM-Cache, and the corresponding experimental setups. Hyperparameter configurations and additional implementation details are presented in Appendix A.3. Comprehensive performance comparisons across benchmarks are reported in Tables 1–4. Ablation studies and a case study are provided in Section 4.3 (e.g., Figure 4) to illustrate the robustness of our method. To facilitate replication, we include the source code and scripts for running experiments in the supplementary materials.

## ETHICS STATEMENT

This work focuses on developing efficient sampling strategies for diffusion-based large language models (dLLMs). Our research does not involve human subjects, personal or sensitive data, or applications directly impacting individuals. All experiments were conducted on publicly available benchmarks, ensuring fairness, reproducibility, and transparency. We release code to promote open research and to enable verification of our results. We are not aware of any potential misuse or harmful applications of the proposed methods beyond standard concerns inherent to language models, such as bias or misuse in downstream tasks, and we encourage responsible use in line with the ICLR Code of Ethics.

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

## A APPENDIX

### A.1 LLM USAGE STATEMENT.

We used a Large Language Model (LLM) solely to aid in polishing the writing and improving the clarity of language. The LLM was not involved in research ideation, experimental design, data analysis, or generation of scientific content. All research ideas, methods, analyses, and conclusions presented in this paper are entirely the work of the authors. The authors take full responsibility for the content of this paper.

### A.2 COMPATIBILITY WITH ADVANCED SAMPLING METHODS.

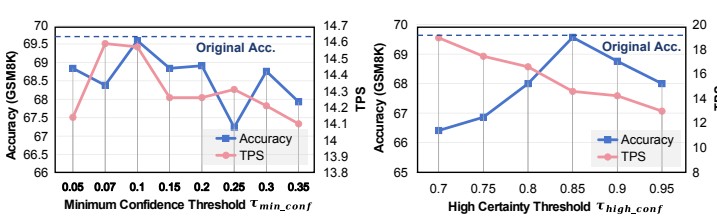

Figure 6: **Effect of Confidence Thresholds on GSM8K.** $\tau_{min\_conf}$ controls the exploratory range, and $\tau_{high\_conf}$ balances accuracy and speed during fast decoding. Other hyperparameters follow the default settings in Section 4.2.

**Sensitivity to Confidence Thresholds.** The core confidence thresholds, $\tau_{min\_conf}$ and $\tau_{high\_conf}$, dictate the behavior of our SlowFast pipeline. As shown in Figure 6, the minimum confidence threshold $\tau_{min\_conf}$ defines the candidate region in the exploratory stage. A moderate value of $\tau_{min\_conf} = 0.1$ is optimal, as lower values lead to unstable regions and higher values create overly conservative, inefficient regions. The high certainty threshold $\tau_{high\_conf}$ directly manages the speed-quality trade-off during the accelerated stage. A higher value leads to more cautious and accurate generation at the cost of speed (TPS). We select $\tau_{high\_conf} = 0.85$, which achieves a near-peak GSM8K score without a drastic reduction in inference speed, striking an effective balance.

### A.3 EXPERIMENTAL DETAILS

To ensure fair and reproducible comparisons, we standardize the setting of inference generation across benchmarks and run each model with its officially recommended settings. The benchmark-specific parameters considered include the number of inference steps, the block length, the total generation length, and the number of few-shot examples. For clarity, we summarize the settings for LLaDA and Dream in Tables 5 and 6, respectively.

Table 5: Experimental settings for LLaDA across benchmarks.

| Task | Steps | Block Length | Generation Length | Few-shot |
|------|-------|--------------|-------------------|----------|
| MMLU | 3 | 3 | 3 | 5 |
| MMLU-pro | 256 | 256 | 256 | 0 |
| GSM8K | 256 | 256 | 256 | 4 |
| Math | 256 | 256 | 256 | 0 |
| GPQA | 256 | 256 | 256 | 5 |
| HumanEval | 256 | 256 | 256 | 0 |
| MBPP | 256 | 256 | 256 | 3 |
| BBH | 256 | 256 | 256 | 3 |

Table 6: Experimental settings for Dream across benchmarks.

| Task | Steps | Generation Length | Few-shot |
|------|-------|-------------------|----------|
| MMLU | 3 | 3 | 5 |
| MMLU-pro | 256 | 256 | 0 |
| GSM8K | 256 | 256 | 8 |
| Math | 256 | 256 | 4 |
| GPQA | 256 | 256 | 5 |
| HumanEval | 256 | 256 | 0 |
| MBPP | 256 | 256 | 3 |
| BBH | 256 | 256 | 3 |

## A.4 ABLATION STUDY ON HYPERPARAMETERS AND ROBUSTNESS.

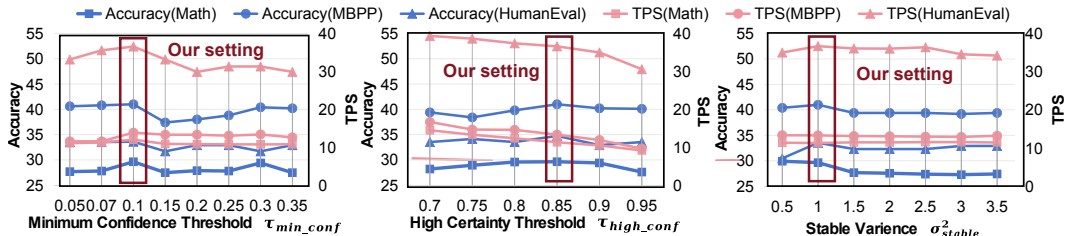

Figure 7: **Parameter Robustness and Sensitivity Across Multiple Benchmarks.** Figure shows sensitivity across MATH, MBPP, and HumanEval. Our setting is shown to be highly robust across all tasks.

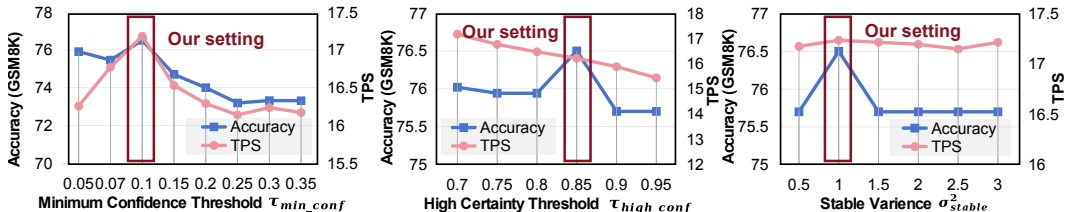

Figure 8: **Parameter Stability Across Model Variants on Dream 7B.** Sensitivity analysis for $\tau_{min\_conf}$, $\tau_{high\_conf}$, and $\sigma^2_{stable}$ was conducted on Dream 7B using the GSM8K benchmark. The findings confirm that the unified set of hyperparameters successfully transfers to this demanding model variant, validating overall parameter stability.

**Hyperparameter Robustness and Generalization Analysis.** The overall hyperparameter analysis confirms the robustness and strong generalization ability of the SlowFast approach. As shown in Figure 7, a single, unified parameter set maintains near-optimal performance across diverse benchmarks (MATH, MBPP, HumanEval). This stability is further validated by Figure 8, where the same unified settings successfully transfer to Dream 7B on the GSM8K benchmark. These findings confirm that the SlowFast approach is highly robust and does not require extensive tuning for new tasks or model variants.

## A.5 PERFORMANCE-EFFICIENCY TRADE-OFF ACROSS STRATEGIES

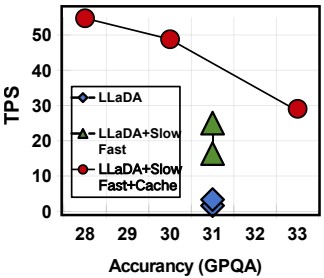

Figure 9: **Performance Efficiency Trade-off on GPQA.**

Figure 9 evaluates the Performance-Efficiency for our method against various baselines under two distinct evaluation setups. The results confirm that the integration of our method consistently shifts the trade-off curve decisively towards the top-right corner (higher TPS and higher Accuracy) compared to the baseline. This superiority holds true across both the standard configuration (5-shot, 256 tokens) and the more demanding scenario (8-shot, 1024 tokens), validating the robustness and scalability of our adaptive sampling strategy.

## A.6 CASE STUDIES ON POSITIONAL PRINCIPLE AND NON-CONTIGUOUS DECODING.

**Robustness to Minor Positional Uncertainty.** Figure 10 confirms SlowFast's ability to cover multiple stable blocks even when separated by unstable regions. Crucially, the distance between the end of one stable block and the start of the next is generally small, allowing our adaptive mechanism to efficiently jump over the intermediate low-confidence span. This ensures that the overall acceleration is maintained without sacrificing accuracy, even when faced with minor positional uncertainties.

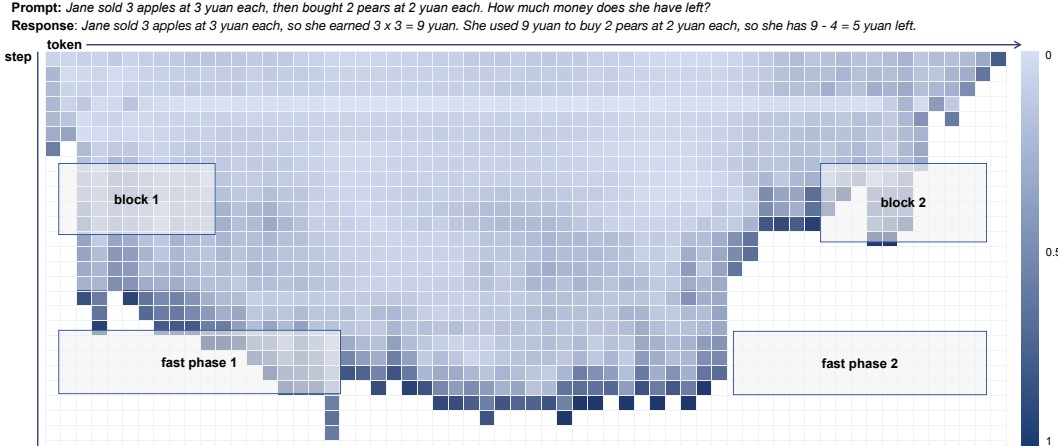

**Prompt:** *Amy sold 4 books at 5 dollars each, then bought 1 pen at 2 dollars each. How much money does she have left?*
**Response**: *Amy sold 4 books at 5 dollars each, which is 4 * 5 = 20 dollars. Then she bought 1 pen at 2 dollars each, which is 1 * 2 = 2 dollars. So, she has 20 - 2 = 18 dollars left.*

Figure 10: **Case Study I: Dynamic Span Coverage During Positional Jumps.** This case study illustrates a scenario where the model exhibits a positional jump between decoded tokens, confirming that the dynamic search window of SlowFast successfully identifies and covers the subsequent stable region.

**Prompt:** *Jane sold 3 apples at 3 yuan each, then bought 2 pears at 2 yuan each. How much money does she have left?*
**Response**: *Jane sold 3 apples at 3 yuan each, so she earned 3 x 3 = 9 yuan. She used 9 yuan to buy 2 pears at 2 yuan each, so she has 9 - 4 = 5 yuan left.*

Figure 11: **Case Study II: SlowFast's Adaptive Response to Induced Non-Contiguous Blocks.** This case study demonstrates the adaptive capability of the SlowFast framework. By manually inducing a positional jump, the figure shows how the dynamic adjustment mechanism immediately recognizes and proceeds to the subsequent high-confidence region.

**Adaptive Handling of Non-Contiguous Blocks.** Figure 11 rigorously tests SlowFast's adaptive limits by manually constructing an extreme scenario featuring two widely separated high-confidence blocks. The visualization confirms that upon encountering the unstable region between the blocks, the dynamic adjustment mechanism successfully identifies the subsequent high-confidence region (Block 2) and resumes acceleration (Fast Phase 2). This successful navigation demonstrates the robustness of SlowFast's dynamic scheduling, confirming its ability to automatically identify and accelerate multiple non-contiguous stable blocks even under extreme conditions.

## A.7 ALGORITHM: SLOWFAST SAMPLING FOR dLLMs

Algorithm 1 outlines the SlowFast Sampling Strategy, a dynamic, two-phase framework for accelerating dLLMs. It starts with the Slow, Exploratory Stage, which proceeds step-by-step (Lines 10-12) to determine a safe decoding span boundary. This boundary is validated by a Variance-based Stability Check (Lines 16-22), confirming stability when the variance of endpoint predictions falls below $\sigma^2_{stable}$. Once stability is confirmed, the algorithm enters the Fast, Accelerated Decoding Stage. Here, acceleration is achieved via In-Span Parallel Decoding (Lines 40-47), where high-confidence tokens within the span are unmasked simultaneously. Additionally, Out-of-Span Caching (Lines 34-39) prepares for the subsequent cycle. This adaptive, two-stage mechanism ensures maximum efficiency is achieved only in reliably stable regions.

---

**Algorithm 1** SlowFast Sampling Strategy

---

**Require:** Masked sequence $y^{(N)}$, Prompt $c$, Max steps $N$, Exploratory limit $K_{max}$, History window $W_{hist}$, Stability threshold $\sigma^2_{stable}$, Confidence thresholds $\tau_{min\_conf}, \tau_{high\_conf}$.
**Ensure:** Generated sequence $y^{(0)}$.
1: $k \leftarrow N$                 ▷ Initialize diffusion step
2: $s_{cycle} \leftarrow 1$           ▷ Start of the current decoding cycle
3: $H_W \leftarrow \emptyset$              ▷ History of convergence points
4: **while** $k > 0$ **and** $s_{cycle} < L$ **do**
5:      **// Phase 1: Exploratory Stage (Slow)**
6:      $step\_count \leftarrow 0$
7:      $is\_stable \leftarrow$ False
8:      **while** $step\_count < K_{max}$ **and** $k > 0$ **do**
9:          Predict $\hat{r}_0^{(k)}$ and confidence $P_\theta(\cdot|c, y^{(k)})$
10:          **Cautious Decoding:**
11:          Unmask Top-$k_{slow}$ tokens in range $[s_{cycle}, L]$
12:          **End Point Prediction:**
13:          $e_{cand}^{(k)} \leftarrow \max\{i \mid i \in [s_{cycle}, L] \wedge P_\theta(\hat{r}_{0,i}^{(k)}) > \tau_{min\_conf}\}$
14:          Update history window $H_W$ with $e_{cand}^{(k)}$
15:          **Stability Check:**
16:          $Var_{curr} \leftarrow \text{Variance}(H_W)$
17:          **if** $|H_W| \geq W_{hist}$ **and** $Var_{curr} < \sigma^2_{stable}$ **then**
18:             $is\_stable \leftarrow$ True
19:             $e_{cycle} \leftarrow \text{Mean}(H_W)$
20:             $k \leftarrow k - 1$
21:             **break**            ▷ Exit slow phase early
22:          **end if**
23:          Update state $y^{(k-1)}$ using standard scheduler
24:          $k \leftarrow k - 1$
25:          $step\_count \leftarrow step\_count + 1$
26:      **end while**
27:      **if not** $is\_stable$ **then**
28:          $e_{cycle} \leftarrow e_{cand}^{(k)}$          ▷ Fallback if no convergence
29:      **end if**
30:      **// Phase 2: Accelerated Decoding Stage (Fast)**
31:      **if** $k > 0$ **then**
32:          Predict $\hat{r}_0^{(k)}$ and confidence $P_\theta(\cdot|c, y^{(k)})$
33:          **Out-of-Span Caching:**
34:          **for** $i \in (e_{cycle}, L]$ **do**
35:             **if** $P_\theta(\hat{r}_{0,i}^{(k)}) < \tau_{min\_conf}$ **then**
36:                Cache features for position $i$
37:             **end if**
38:          **end for**
39:          **In-Span Parallel Decoding:**
40:          $S_{high} \leftarrow \{i \mid i \in [s_{cycle}, e_{cycle}] \wedge P_\theta(\hat{r}_{0,i}^{(k)}) > \tau_{high\_conf}\}$
41:          **if** $S_{high} \neq \emptyset$ **then**
42:             $y_i^{(k-1)} \leftarrow \hat{r}_{0,i}^{(k)}$ for all $i \in S_{high}$     ▷ Aggressive unmasking
43:          **else**
44:             Unmask Top-$k_{fast}$ tokens in $[s_{cycle}, e_{cycle}]$     ▷ Fallback refinement
45:          **end if**
46:          $s_{cycle} \leftarrow e_{cycle}$          ▷ Advance the stable region start
47:          $k \leftarrow k - 1$
48:      **end if**
49: **end while**
50: **return** $y^{(0)}$

---

