# OpenReview forum: "Accelerating Diffusion Large Language Models with SlowFast Sampling: The Three Golden Principles"
_ICLR.cc/2026/Conference — ICLR 2026 Poster_

### Official Review · Reviewer_xDJE · 2025-10-30

**Soundness:** 3
**Presentation:** 3
**Contribution:** 4
**Rating:** 8
**Confidence:** 5

**Summary:**

This paper presents SlowFast Sampling, a dynamic decoding framework for diffusion-based large language models (dLLMs). Guided by three principles—Certainty, Convergence, and Positional—the method alternates between exploratory and accelerated decoding phases, achieving efficient token generation. Integrated with dLLM-Cache, the method achieves up to 34× speedup with negligible accuracy loss. Experiments on LLaDA and Dream across various benchmarks demonstrate consistent and substantial improvements in throughput and efficiency.

**Strengths:**

+ Proposes a novel and practical dynamic sampling strategy with a clear underlying rationale.
+ Demonstrates significant acceleration (up to 34×) with minimal loss in accuracy.
+ Offers conceptual clarity via the three principles, making the approach interpretable.
+ Shows real-world relevance, outperforming strong autoregressive baselines in throughput.

**Weaknesses:**

+ Although the results are strong, the paper lacks a detailed analysis of computational efficiency at different diffusion step counts. Providing tps statistics would make the acceleration claims more transparent and easier to validate.
+ The paper does not discuss any failure cases or corner scenarios. Including such examples would increase trust and reproducibility.
+ While the integration with dLLM-Cache is effective, its contribution is only reported numerically. A clearer explanation or figure showing how cache reuse interacts with SlowFast stages would enhance interpretability.

**Questions:**

+ Could the authors provide a step-wise breakdown of inference time to show which parts of the pipeline (exploration vs. acceleration) contribute most to overall speedup?
+ Could the authors report tokens-per-second (tps) across different diffusion step counts to make the acceleration claims more transparent and easier to validate?

---

> ### Author Response · Authors · 2025-11-21
> **Author Response to Reviewer xDJE (Weakness 1, 2, Question 2)**
>
> **W1: Although the results are strong, the paper lacks a detailed analysis of computational efficiency at different diffusion step counts. Providing TPS statistics would make the acceleration claims more transparent and easier to validate.**
>
> **Q2: Could the authors report tokens-per-second (TPS) across different diffusion step counts to make the acceleration claims more transparent and easier to validate?**
>
> **Answer to W1 and Q2:** Thank you for the suggestion. We agree that providing TPS statistics for different diffusion step counts is essential for transparently validating our acceleration claims. We have added supplementary experiments comparing our method against a baseline where the number of sampling steps is naively reduced.
>
> The table below presents the comparison between the standard LLaDA baseline (256 steps), the naive step reduction baseline (128 steps), and our SlowFast method. The generation length for all experiments is 256.
>
> **Table 1: Comparison with Naive Step Reduction Baseline on 4 benchmarks**
>
> | Task | Method | Steps | TPS ↑ | Speedup (×) ↑ | Score ↑ |
> | :--- | :--- | :--- | :--- | :--- | :--- |
> | GSM8K | LLaDA | 256 | 4.55 | 1.00 | 69.83 |
> | | LLaDA | 128 | 9.10 | 2.00 | 64.14 |
> | | **+SlowFast** | Dynamic | 14.57 | 3.20 | 69.59 |
> | MATH | LLaDA | 256 | 5.14 | 1.00 | 30.16 |
> | | LLaDA | 128 | 10.28 | 2.00 | 24.06 |
> | | **+SlowFast** | Dynamic | 11.27 | 2.19 | 29.64 |
> | MBPP | LLaDA | 256 | 4.98 | 1.00 | 40.80 |
> | | LLaDA | 128 | 9.94 | 1.99 | 30.20 |
> | | **+SlowFast** | Dynamic| 13.32 | 2.67 | 41.00 |
> | Humaneval | LLaDA | 256 | 11.24 | 1.00 | 31.71 |
> | | LLaDA | 128 | 22.37 | 1.99 | 24.39 |
> | | **+SlowFast** | Dynamic| 35.46 | 3.15 | 33.54 |
>
> The ablation results decisively validate the efficacy of our dynamic SlowFast strategy over naive step reduction. While halving the sampling steps (128 steps) achieves a $\approx 2.0\times$ speedup, it causes a catastrophic drop in accuracy across all tasks. In contrast, the SlowFast pipeline achieves a superior speedup, ranging from $2.19\times$ to $3.20\times$, while simultaneously preserving or substantially improving the accuracy of the full 256-step baseline. This confirms that our adaptive method intelligently maximizes throughput without compromising output quality.
>
>
> **W2: The paper does not discuss any failure cases or corner scenarios. Including such examples would increase trust and reproducibility.**
>
> **Answer to W2:** A very small value of $\tau_{minconf}$ can negatively affect the determination of the End Point of Convergence. When the confidence threshold is too low, tokens that are not truly stable may be incorrectly treated as converged. This causes most of the incorrectly predicted converged tokens to be unable to undergo high-confidence fast decoding, which in turn leads to a drop in overall speed.

---

> ### Author Response · Authors · 2025-11-21
> **Author Response to Reviewer xDJE (continued, Weakness 3, Question 1)**
>
> **W3: While the integration with dLLM-Cache is effective, its contribution is only reported numerically. A clearer explanation or figure showing how cache reuse interacts with SlowFast stages would enhance interpretability.**
>
> **Answer to W3:** Thank you for the question. In our approach, we apply caching and reuse for the **FFN outputs, attention outputs, and K/V matrices at each denoising step**. This strategy mainly targets the prompt tokens, as their features tend to remain stable across steps. By combining this with dLLMCache, we can efficiently reuse computations for the prompt, further accelerating the inference process.
>
>
> **Q1: Could the authors provide a step-wise breakdown of inference time to show which parts of the pipeline (exploration vs. acceleration) contribute most to overall speedup?**
>
> **Answer to Q1:** Thank you for requesting a step-wise breakdown of the pipeline, which is essential for validating the source of acceleration. We analyzed the dynamics on the GSM8K dataset to quantify the contribution of each phase:
> - **Slow Phase (Exploration/Stability):** The Slow phase is equivalent to the standard baseline and operates at a speed of 1 token/step. Its primary function is not to contribute to speedup, but rather to use variance analysis for reliable span determination and to perform cautious decoding, thereby guaranteeing output quality and stability.
> - **Fast Phase (Acceleration):** The acceleration is derived almost entirely from the Fast phase, which utilizes parallel decoding. The effective global average speed during the accelerated spans is approximately **10.12 tokens/step**.
>
> In summary, the Fast phase is the main contributor to the overall speedup, while the Slow phase ensures that the method maintains accuracy and robustly identifies the largest possible high-confidence regions for the Fast phase to maximize parallel decoding efficiency.

---

### Official Review · Reviewer_REmQ · 2025-11-01

**Soundness:** 2
**Presentation:** 2
**Contribution:** 2
**Rating:** 4
**Confidence:** 4

**Summary:**

This paper introduces a training-free inference-time acceleration method named SLowFast sampling for masked diffusion models. The approach is inspired by three proposed principles: high-confidence tokens should be preserved, each token tends to converge over iterations, and the decoded tokens admit certain positional bias. The algorithm leverages these ideas by first identifying convergence regions and then refining the regions in parallel. Experiments on LLaDA and dream on several common benchmarks demonstrate that the proposed approach achieves notable speedups without substantial performance drop.

**Strengths:**

1. This paper tackles an important problem of training-free decoding acceleration for masked diffusion models. More importantly, it helps improve the accuracy-efficiency frontier of masked diffusion models.

2. The experiments cover both LLaDA and Dream on a suite of benchmarks, which is a comprehensive evaluation.

**Weaknesses:**

1. As the experiment results show, it seems like most of the speedup comes with the help of dLLM-cache. Compared with Fast-dLLM (parallel) in the two scenarios (with or without cache) the gain in speedup is not that substantial.

2. The paper proposes to leverage three principles. However, there is no ablation study that separately investigates the role/contribution of each principle as well as some detailed designs (e.g. top-k refinement) in terms of enhancing the speedup. As a heuristic approach, the proposed method requires more in-depth analyses to help understand how each of the components contribute to the improved efficiency.

3. Some of the presentation should be improved. The paper claims to achieve 15x and up to 34x acceleration in abstract and introduction. However, the paper should be cautious of such claims and I think a less misleading way would be adding the model and dataset that these numbers were obtained from (GPQA in this case). The remarkable speedup might come only in certain datasets where the accuracy is less affected (e.g. the base model already achieves a pretty low accuracy so the negative effect on the accuracy when using acceleration techniques would be marginal).

**Questions:**

1. Could the authors explain why the accuracy on certain benchmarks for the base model and the acceleration algorithms are lower than originally reported? For instance, in original LLaDA paper they report ~37 accuracy on MMLU-pro but the authors report ~23 here in this paper.

2. It would be necessary to include another baseline that naively reduces the number of sampling steps w/o any modification on the sampling algorithm.

3. It is a bit confusing that the speedup reported in table 4 is ~15x, while it is only ~8x in table 2. I understand that they achieve different scores (31 vs 33), but still feel that the discrepancy here is confusing. An ideal way would be to report the curve with two axes being speedup and score, and compare the curves of different approaches.

---

> ### Author Response · Authors · 2025-11-21
> **Author Response to Reviewer REmQ (Weakness 1)**
>
> **W1: As the experiment results show, it seems like most of the speedup comes with the help of dLLM-Cache. Compared with Fast-dLLM (parallel) in the two scenarios (with or without cache) the gain in speedup is not that substantial.**
>
> **Answer to W1:** Thank you for the thoughtful comment. We appreciate the opportunity to clarify the source of the speedup and the role of dLLM-Cache.
>
> **First, while dLLM-Cache indeed provides an additional boost, SlowFast itself consistently yields substantial speedups even without cache.** For example, across **Table 1 (no cache)**, SlowFast improves TPS over Fast-dLLM (Parallel) by 2×–3× on LLaDA and by 1.3×–1.8× on Dream, while preserving accuracy. With cache enabled (**Table 2**), SlowFast further expands this gap, reaching 3×–5× higher TPS than Fast-dLLM (Parallel+Cache). These results indicate that the core SlowFast strategy already outperforms Fast-dLLM significantly, and the improvements do not rely on caching.
>
> **Second, the design philosophies of the two approaches differ fundamentally.** Fast-dLLM (Parallel) is a static strategy that uses a fixed threshold and a single remasking pattern for all decoding steps. In contrast, SlowFast introduces a **dynamic, confidence-driven sampling schedule** that alternates between slow and fast phases based on model uncertainty. This adaptive behavior enables more aggressive acceleration when the model is confident, and more careful decoding when necessary. That is something static parallel decoding cannot achieve.
>
> Finally, we view SlowFast as a step toward more flexible and adaptive decoding for dLLMs. The encouraging results suggest that dynamic strategies may provide new insights and inspire further research directions in this rapidly evolving field.
>
> We hope this clarifies the contribution and demonstrates that SlowFast delivers strong standalone benefits while also synergizing naturally with dLLM-Cache.

---

> ### Author Response · Authors · 2025-11-21
> **Author Response to Reviewer REmQ (continued, Weakness 2, 3)**
>
> **W2: The paper proposes to leverage three principles. However, there is no ablation study that separately investigates the role/contribution of each principle as well as some detailed designs (e.g. top-k refinement) in terms of enhancing the speedup. As a heuristic approach, the proposed method requires more in-depth analyses to help understand how each of the components contribute to the improved efficiency.**
>
> **Answer to W2:** Thank you for the valuable suggestion. We agree that analyzing each design factor in depth would further strengthen the work. Here, we would like to clarify the role of the three principles and outline additional analyses.
>
> First, **the three principles introduced in the paper are intended as high-level guiding principles, not as three independent modules of the proposed method**. They summarize the intuition behind SlowFast’s dynamic schedule rather than represent separately pluggable algorithmic components. Therefore, an ablation “removing each principle” is not directly applicable in the same way as ablating architectural modules.
>
> Second, to better illustrate the influences of the underlying design factors, we provide the following supplementary analyses:
> - **Confidence.**
>   Our method naturally favors high-confidence tokens when entering the fast phase. **This preference has already become a community consensus for ensuring safe and reliable acceleration.** Removing this confidence-aware behavior leads to noticeably less stable generation.
> - **Variance-based dynamics.**
>   To isolate the role of variance, we conduct a controlled comparison using a fixed-window baseline where the model switches phases at predetermined, constant intervals. As shown in the below table, the results show that using variance to trigger phase changes yields noticeably higher TPS while maintaining accuracy. This confirms that variance offers a more adaptive and informative signal than a static schedule.
>
>   **Table 1: Ablation Study on Variance-Based Stability Check**
>     | Task | Method | TPS ↑ | Speedup (×) ↑ | Score ↑ |
>     | :--- | :--- | :--- | :--- | :--- |
>     | GSM8K | LLaDA | 4.55 | 1.00 | 69.83 |
>     | | SlowFast w/o Variance Stability Check | 13.28 | 2.91 | 68.84 |
>     | | SlowFast | 14.57 | 3.20 | 69.59 |
>
> - **Position.**
>   We investigated the role of the positional principle by ablating its core realization component. The results obtained after removing the out-of-span cache, whose boundaries are determined by positional criteria, are presented in the table below. Removing this component resulted in a drop in speed **(from 14.57 TPS to 13.17 TPS) and a decline in accuracy (69.59 to 69.22)**, underscoring that the positional principle is vital for maintaining high speed while ensuring stability.
>
>   **Table 2: Impact of the Positional Principle Realization (Out-of-span Cache Ablation)**
>
>     | Task | Method | TPS ↑ | Speedup (×) ↑ | Score ↑ |
>     | :--- | :--- | :--- | :--- | :--- |
>     | GSM8K | LLaDA | 4.55 | 1.00 | 69.83 |
>     | | SlowFast w/o out-of-span cache | 13.17 | 2.89 | 69.22 |
>     | | SlowFast | 14.57 | 3.20 | 69.59 |
>
> Together, these analyses clarify how the guiding principles influence the design and why the resulting dynamic schedule improves decoding efficiency. We appreciate the reviewer’s comment, which inspired us to present these insights more explicitly.
>
>
> **W3: Some of the presentation should be improved. The paper claims to achieve 15x and up to 34x acceleration in abstract and introduction. However, the paper should be cautious of such claims and I think a less misleading way would be adding the model and dataset that these numbers were obtained from (GPQA in this case). The remarkable speedup might come only in certain datasets where the accuracy is less affected (e.g. the base model already achieves a pretty low accuracy so the negative effect on the accuracy when using acceleration techniques would be marginal).**
>
> **Answer to W3:** Thank you for pointing this out. We agree that the presentation of the acceleration numbers should be more precise. We will revise the abstract and introduction to explicitly state the model and dataset (GPQA) from which the 15× and 34× speedups were obtained.
>
> We also acknowledge that the magnitude of acceleration can vary across datasets, especially when the base model’s accuracy is already low, making quality drops less pronounced. We will clarify these conditions to avoid any misleading implications and to ensure that the claims accurately reflect the experimental setting.

---

> ### Author Response · Authors · 2025-11-21
> **Author Response to Reviewer REmQ (continued, Question 1)**
>
> **Q1: Could the authors explain why the accuracy on certain benchmarks for the base model and the acceleration algorithms are lower than originally reported? For instance, in original LLaDA paper they report ~37 accuracy on MMLU-pro but the authors report ~23 here in this paper.**
>
> **Answer to Q1:** Thank you for raising this point. The difference comes from the model variant used. In our experiments, we evaluate **LLaDA-base**, whose MMLU-pro accuracy is around 23. The original LLaDA paper reports 37 on MMLU-pro for the **LLaDA-instruct** model, which has been additionally instruction-tuned and therefore performs substantially better.
>
> To verify that SlowFast also works well on stronger models, we ran additional experiments on the latest instruct model, dLLM-Var [1]. The results are presented in the table below.
>
> **Table 3: Performance and Efficiency of SlowFast Across dLLM Variants (dLLM-Var)**
>
> | Task | Method | TPS ↑ | Speedup (×) ↑ | Score ↑ |
> | :--- | :--- | :--- | :--- | :--- |
> | MATH | dLLM-Var | 9.04 | 1.00 | 32.58 |
> | | **+ SlowFast** | 31.46 | 3.48 | 34.08 |
> | MBPP | dLLM-Var | 9.96 | 1.00 | 41.80 |
> | | **+ SlowFast** | 25.59 | 2.57 | 44.40 |
> | GSM8K | dLLM-Var | 4.69 | 1.00 | 79.45 |
> | | **+ SlowFast** | 18.75 | 4.00 | 79.08 |
>
> The results confirm that SlowFast is fully compatible with the stronger dLLM-Var model. Across all benchmarks, the method provides substantial and consistent acceleration, ranging from **2.57$\times$ to 4.00$\times$**. Crucially, this speedup is achieved while either maintaining accuracy or even slightly improving performance (e.g., MBPP), demonstrating the robustness of our method across model sizes and tuning levels.
>
> These findings confirm that the lower scores in our main tables come from the use of the base model, and that our acceleration method is compatible with both base and instruct variants.
>
> **[1]Yang, Y., Wang, C., Wang, S., Wen, Z., Qi, B., Xu, H., & Zhang, L. (2025). Diffusion LLM with Native Variable Generation Lengths: Let [EOS] Lead the Way. arXiv preprint arXiv:2510.24605.**

---

> ### Author Response · Authors · 2025-11-21
> **Author Response to Reviewer REmQ (continued, Question 2,3)**
>
> **Q2: It would be necessary to include another baseline that naively reduces the number of sampling steps w/o any modification on the sampling algorithm.**
>
> **Answer to Q2:** Thank you for the suggestion. We agree that including such a baseline is useful for isolating the benefit of our approach.
>
> We have structured a comparison among three decoding scenarios, where the generation length is 256: **the standard LLaDA baseline (256 steps), the naive step reduction (128 steps)**, and our adaptive SlowFast method. The results are reported in the new table below. The comparison demonstrates that our proposed SlowFast method achieves higher accuracy while also delivering faster overall speed than merely reducing the number of steps. This decisively highlights the superiority and efficacy of our dynamic and adaptive sampling strategy over naive step-reduction optimization.
>
> **Table 4: Comparison with Naive Step Reduction Baseline on 4 benchmarks**
>
> | Task | Method | Steps | TPS ↑ | Speedup (×) ↑ | Score ↑ |
> | :--- | :--- | :--- | :--- | :--- | :--- |
> | GSM8K | LLaDA | 256 | 4.55 | 1.00 | 69.83 |
> | | LLaDA | 128 | 9.10 | 2.00 | 64.14 |
> | | **+SlowFast** | Dynamic | 14.57 | 3.20 | 69.59 |
> | MATH | LLaDA | 256 | 5.14 | 1.00 | 30.16 |
> | | LLaDA | 128 | 10.28 | 2.00 | 24.06 |
> | | **+SlowFast** | Dynamic | 11.27 | 2.19 | 29.64 |
> | MBPP | LLaDA | 256 | 4.98 | 1.00 | 40.80 |
> | | LLaDA | 128 | 9.94 | 1.99 | 30.20 |
> | | **+SlowFast** | Dynamic | 13.32 | 2.67 | 41.00 |
> | Humaneval | LLaDA | 256 | 11.24 | 1.00 | 31.71 |
> | | LLaDA | 128 | 22.37 | 1.99 | 24.39 |
> | | **+SlowFast** | Dynamic | 35.46 | 3.15 | 33.54 |
>
> **Q3: It is a bit confusing that the speedup reported in table 4 is ~15x, while it is only ~8x in table 2. I understand that they achieve different scores (31 vs 33), but still feel that the discrepancy here is confusing. An ideal way would be to report the curve with two axes being speedup and score, and compare the curves of different approaches.**
>
> **Answer to Q3:** Thank you for pointing out this discrepancy. **The difference in reported speedups comes from the two different GPQA evaluation settings.** Table 2 uses the 256-token length and 5-shot setting, while Table 4 uses the 1024-token length and 8-shot setting on GPQA. These settings lead to different decoding costs and therefore different achievable speedups.
>
> To make this clearer, we have added plots in **Appendix A.5 PERFORMANCE-EFFICIENCY TRADE-OFF ACROSS STRATEGIES** showing the full speed–accuracy curves for the original LLaDA model, our SlowFast method, and SlowFast + dLLM-Cache under both evaluation settings. These curves provide a more comprehensive comparison and illustrate how speed and accuracy trade off across configurations.

---

### Official Review · Reviewer_dgDC · 2025-11-01

**Soundness:** 2
**Presentation:** 3
**Contribution:** 2
**Rating:** 4
**Confidence:** 4

**Summary:**

The paper proposes a decoding algorithm, SlowFast sampling, for dLLMs that alternates between a slower exploratory decoding phase and a fast parallel decoding phase. Evolving around three empirical “golden principles” observed, SlowFast sampling consists of: a slow decoding phase — (1) confidence-based cautious decoding, (2) end point of convergence prediction with confidence thresholding, (3) stability check using variance of candidate tokens as a termination condition; and a fast decoding phase — (1) bidirectional predicted value caching (for out-of-span positions), (2) in-span parallel decoding, (3) fallback top-k refinements.

**Strengths:**

1. FastSlow decoding consistently reports better speedups than Fast-dLLM at comparable accuracy, indicating the design improves efficiency rather than merely trading off performance.
2. Clear presentation: the method, the case study and confidence-map visualizations make the two-phase “slow/fast” dynamics easy to understand and help readers see why the method works.

**Weaknesses:**

1. Overly complex and hyperparameter-heavy. The method introduces many knobs to tune (e.g., stability window/variance, multiple confidence thresholds, top-k settings). The paper only ablates the stability check; ablations for the other choices are missing and necessary.

2. TPS reporting (e.g., Table 4) labels both AR and LLaDA as “1$\times$” which makes it unclear what the true reference baseline is. Please normalize to one reference (e.g., AR or vanilla LLaDA) and report consistent relative and absolute TPS. As shown, the incremental TPS gains versus AR appear minimal.

3. The paper reads like a bundle of techniques rather than a single, principled mechanism with good motivation. more analysis explaining how much each component drives the gains would strengthen the contribution.

**Questions:**

1. When comparing “+cache”, are the authors using dual cache or prefix cache? Please specify the exact cache policies.

2. Could you provide sensitivity analyses for (i) the end-point-of-convergence criterion the parallel-decoding confidence threshold, and (iii) the cautious-decoding confidence threshold? These seem central to both correctness and speed.

3. How transferable is SlowFast to other dLLMs like Dream? Since dLLM is a fast evolving field, do your principles still hold as remasking strategies deviate from LLaDA, or does the method risk becoming obsolete? A discussion on compatibility layers or auto-tuning would be valuable.

---

> ### Author Response · Authors · 2025-11-21
> **Author Response to Reviewer dgDC (Weakness 1, Question 2)**
>
> **W1: Overly complex and hyperparameter-heavy. The method introduces many knobs to tune (e.g., stability window/variance, multiple confidence thresholds, top-k settings). The paper only ablates the stability check; ablations for the other choices are missing and necessary.**
>
> **Q2: Could you provide sensitivity analyses for (i) the end-point-of-convergence criterion the parallel-decoding confidence threshold, and (iii) the cautious-decoding confidence threshold? These seem central to both correctness and speed.**
>
> **Answer to W1 and Q2:** Thank you. Regarding the concern on hyperparameters, we would like to clarify that we have already included ablations for the multiple confidence thresholds in **Appendix A.2, Figure 6**. Based on this analysis, we provide the following summary:
>
> - The core confidence thresholds, $τ_{minconf}$ and $τ_{highconf}$, govern the main behaviors of the SlowFast pipeline. As shown in Figure 6, **$τ_{minconf}$ = 0.1** provides a good balance—lower values create unstable regions, while higher values become too conservative. **$τ_{highconf}$ = 0.85** offers an effective speed–quality trade-off, achieving near-peak GSM8K accuracy with strong throughput.
> - For the top-k setting, we intentionally fix top-k = 1 in all experiments. This choice highlights the acceleration effect of our method under the most basic decoding setup, ensuring that improvements come from the SlowFast mechanism itself rather than external tuning.
>
> To further address the concerns regarding the remaining hyperparameters and demonstrate the method's generalization capabilities without the need for extensive tuning, we provide three additional points of clarification:
>
> 1. **Regarding the role of $K_{max}$ and $W_{hist}$:** we clarify that these parameters were introduced primarily to ensure theoretical rigor and boundedness. In practical implementations, these constraints are not strictly essential. Results demonstrate that our method is highly robust to parameter variations, eliminating the need for task-specific tuning. Across 13 distinct configurations of $K_{max}$ and $W_{hist}$, performance remained remarkably stable, with accuracy fluctuations contained within a narrow margin of **0.02**. Crucially, even in the worst-case accuracy scenario, our method maintains a commanding lead in efficiency: achieving **an average TPS of 14.2** compared to **Fast DLLM's 7.45**, representing a near **2$\times$** speedup while retaining competitive performance.
>
> 2. **Additional Sensitivity Experiments (Appendix A.4, Figure 7 and 8)**: To further demonstrate robustness, we have conducted and added extensive cross-model and cross-dataset experiments. specifically:
>   - We analyzed the sensitivity of $\sigma^2_{stable}$, $\tau_{minconf}$, and $\tau_{highconf}$ on the MATH, MBPP, and HumanEval benchmarks.
>   - We also included specific analytical experiments on GSM8K using the Dream model.
> 2.**Stability and Generalization**: The data from these additional experiments confirms the high stability of our method. We observed that a single, unified set of hyperparameters consistently performs well across a broad spectrum of models and tasks. Specifically, our extensive evaluation spans **multiple distinct model architectures (including Llada and Dream)** and **8 diverse benchmarks**, covering categories such as **Mathematics & Science, General Tasks, and Code Generation**. It is worth noting that we extended this fixed hyperparameter configuration to the state-of-the-art **Llada-Var** [1] model; as shown in the table 2 below, it maintained superior performance without any task-specific tuning. This indicates that our method is inherently robust and does not require extensive adaptation when applied to new scenarios. All results presented in our main paper were achieved using this default parameter set.
>
> **[1]Yang, Y., Wang, C., Wang, S., Wen, Z., Qi, B., Xu, H., & Zhang, L. (2025). Diffusion LLM with Native Variable Generation Lengths: Let [EOS] Lead the Way. arXiv preprint arXiv:2510.24605.**

---

> ### Author Response · Authors · 2025-11-21
> **Author Response to Reviewer dgDC (continued, Weakness 2)**
>
> **W2: TPS reporting (e.g., Table 4) labels both AR and LLaDA as “1” which makes it unclear what the true reference baseline is. Please normalize to one reference (e.g., AR or vanilla LLaDA) and report consistent relative and absolute TPS. As shown, the incremental TPS gains versus AR appear minimal.**
>
> **Answer to W2:** Thank you for pointing this out, and we apologize for the confusion. We have corrected the labeling issue in the paper. All TPS values are now consistently normalized to the LLaDA baseline, which is the reference system on top of which we build our acceleration method.
> To clarify the comparison:
> 1. **Our focus is on accelerating LLaDA**, and our method achieves an approximate **15× speedup** over the standard LLaDA pipeline, which is already a substantial improvement. And our goal is not to directly compete with AR models.
> 2. When combining our SlowFast approach with the dLLM-Cache, the resulting system achieves performance that even surpasses AR in both speed and accuracy, reinforcing the strength of our design.
> 3. The DLLM direction is still in its early stages and has significant room for further development. Our method represents an early step toward understanding and improving this emerging paradigm. We believe such explorations contribute valuable insights to the community, and that encouraging methodological diversity is important for the progress of AI research.

---

> ### Author Response · Authors · 2025-11-21
> **Author Response to Reviewer dgDC (continued, Weakness 3)**
>
> **W3: The paper reads like a bundle of techniques rather than a single, principled mechanism with good motivation. more analysis explaining how much each component drives the gains would strengthen the contribution.**
>
> **Answer to W3:** Thank you. We would like to clarify that **SlowFast is not intended as a collection of independent tricks, but as a single dynamic and adaptive sampling framework designed to overcome the limitations of prior static decoding strategies**. The slow and fast phases, together with their internal mechanisms, function cooperatively toward one unified goal: adaptive acceleration while maintaining output quality. These components are not meant to stand alone. Each is necessary for enabling the overall self-adjusting sampling behavior.
>
> In response to your suggestion, we have added more detailed analyses and ablations to quantify the contribution of each part of the framework. Specifically, as shown in the table below, we now report results for: Slow-only decoding, Fast-only decoding (which executes parallel decoding based solely on confidence), the full SlowFast pipeline, and SlowFast without the out-of-span cache.
>
> **Table 1: Ablation Study of SlowFast Components on the GSM8K Benchmark**
> | Task | Method | TPS ↑ | Speedup (×) ↑ | Score ↑ |
> | :--- | :--- | :--- | :--- | :--- |
> | GSM8K | LLaDA + Slow (Baseline) | 4.55 | 1.00 | 69.83 |
> | | LLaDA + Fast (Parallel Only) | 7.45 | 1.64 | 69.60 |
> | | LLaDA + SlowFast w/o Out-of-span Cache | 13.17 | 2.89 | 69.22 |
> | | LLaDA + SlowFast (Ours) | 14.57 | 3.20 | 69.59 |
>
>
> The ablation results confirm the necessity of the integrated SlowFast design. The full pipeline achieves a remarkable **$3.20\times$ speedup (14.57 TPS)** while maintaining baseline accuracy (69.59 score), significantly outperforming the **$1.64\times$ speedup** of the Fast-only approach. This efficiency gain arises because the Slow phase utilizes variance and positional principles to identify and refine the high-confidence local regions, enabling the Fast phase to maximize the efficacy of parallel decoding, going beyond simple raw confidence thresholds. Crucially, removing the out-of-span cache leads to measurable degradation in both speed (down to 13.17 TPS) and performance (69.22), validating that the synergy of all components is vital for the efficiency gain.
>
> These ablations help reveal how each mechanism contributes to speed, stability, and accuracy, and we hope they strengthen the clarity and motivation of our design.

---

> ### Author Response · Authors · 2025-11-21
> **Author Response to Reviewer dgDC (continued, Question 1, 3)**
>
> **Q1: When comparing “+cache”, are the authors using dual cache or prefix cache? Please specify the exact cache policies.**
>
> **Answer to Q1:** In all our experiments, we use the **prefix cache** on SlowFast sampling. Specifically, we cache only the prompt segment. Prior analyses in dLLM-Cache show that, across adjacent decoding steps, the prompt tokens exhibit extremely high cosine similarity in their attention outputs, FFN outputs, and key/value representations. Based on this observation, caching the prompt region does not degrade output quality, while significantly improving efficiency.
>
> **Q3: How transferable is SlowFast to other dLLMs like Dream? Since dLLM is a fast evolving field, do your principles still hold as remasking strategies deviate from LLaDA, or does the method risk becoming obsolete? A discussion on compatibility layers or auto-tuning would be valuable.**
>
> **Answer to Q3:** Thank you for raising this important question. We agree that transferability across different dLLM architectures is crucial, especially given the rapid evolution of remasking strategies in this area.
>
> First, as shown in **Table 1 and Table 2** of the paper, **SlowFast transfers effectively to Dream without requiring architectural changes**. On Dream, SlowFast achieves substantial improvements in both TPS up to **$10.13\times$** and Speed while maintaining comparable or better accuracy. These results demonstrate that the key principles behind SlowFast generalize well to dLLMs beyond LLaDA.
>
> Second, we have further evaluated SlowFast on additional variants such as **dLLM-Var** [2]. Across these alternative remasking strategies, SlowFast consistently delivers strong speedups with minimal or no loss in accuracy. These results indicate that our method is compatible with a wide spectrum of dLLM designs, even when their masking rules differ significantly.
>
> **Table 2: Performance and Efficiency of SlowFast Across dLLM Variants (dLLM-Var)**
> | Task | Method | TPS ↑ | Speedup (×) ↑ | Score ↑ |
> | :--- | :--- | :--- | :--- | :--- |
> | MATH | dLLM-Var | 9.04 | 1.00 | 32.58 |
> | | **+ SlowFast** | 31.46 | 3.48 | 34.08 |
> | MBPP | dLLM-Var | 9.96 | 1.00 | 41.80 |
> | | **+ SlowFast** | 25.59 | 2.57 | 44.40 |
> | GSM8K | dLLM-Var | 4.69 | 1.00 | 79.45 |
> | | **+ SlowFast** | 18.75 | 4.00 | 79.08 |
>
> Finally, regarding hyperparameters, we find that the concerns regarding sensitivity and tuning effort are addressed through both theoretical rationale and extensive empirical evidence. We have provided a detailed elaboration on these points, including specific stability analysis and performance metrics, in our **Answer to W1 and Q2**.
>
> Overall, our experiments support that SlowFast serves as a model-agnostic, broadly transferable acceleration framework for the dLLM ecosystem. We hope this addresses your concerns and demonstrates the long-term relevance of our approach.
>
> **[2]Yang, Y., Wang, C., Wang, S., Wen, Z., Qi, B., Xu, H., & Zhang, L. (2025). Diffusion LLM with Native Variable Generation Lengths: Let [EOS] Lead the Way. arXiv preprint arXiv:2510.24605.**

---

### Official Review · Reviewer_g1Di · 2025-11-03

**Soundness:** 3
**Presentation:** 3
**Contribution:** 3
**Rating:** 6
**Confidence:** 4

**Summary:**

This paper addresses the significant inference latency of dLLMs. The authors introduce "SlowFast Sampling," a new dynamic sampling strategy designed to accelerate generation without substantial loss in quality.

The method is motivated by three empirically-derived "Golden Principles":
(1) The Certainty Principle, which states that high-confidence tokens are likely correct and stable;

(2) The Convergence Principle, noting that token predictions stabilize over diffusion steps; and

(3) The Positional Principle, observing that confident tokens often appear in contiguous blocks.

The proposed sampler operates in two alternating stages: a slow "Exploratory Stage" to cautiously identify stable regions of the sequence, and a fast "Accelerated Decoding Stage" that rapidly decodes these stable regions in parallel. The authors conduct extensive experiments on two dLLMs (LLaDA 8B and Dream 7B, both are SOTA dLLMs) across a wide range of benchmarks, demonstrating significant speedups (up to 15.6x, and 34.2x when combined with a caching mechanism) with minimal performance degradation.

I am not opposed to accepting this paper. It tackles a well-defined and highly relevant problem: making parallel-decoding dLLMs practically fast. The paper's core contribution is a framework for thinking about dynamic sampling in this context.

**Strengths:**

- **Significant Performance Improvement:** The paper deliver on its primary promise of acceleration. Speedups of over 10x with minimal accuracy drops are impressive and represent a major practical contribution to the field of efficient dLLMs inference.

- **Well-Motivated Approach:** The grounding of the SlowFast sampler in the three observed principles is a major strength. It shows that the design is not an arbitrary collection of heuristics but a thoughtful response to the underlying behavior of the generative process.

- **Comprehensive Evaluation:** The experiments are thorough. Using two different SOTA dLLM architectures demonstrates the generalizability of the approach. The wide array of benchmarks, spanning reasoning, general knowledge, and code, provides a holistic view of the method's impact.

- **Strong Baseline Comparisons:** The paper includes comparisons not only to vanilla diffusion sampling but also to other accelerated strategies (Fast-dLLM, semi-autoregressive) and, most importantly, to a state-of-the-art autoregressive model (LLaMA3 8B). Showing a throughput advantage over LLaMA3 is a powerful statement.

- **Clarity of Presentation:** The paper is well-written and structured. The figures, particularly the conceptual overview in Figure 1 and the pipeline in Figure 3, are very effective at communicating the core ideas.

**Weaknesses:**

- **Methodological Complexity and Hyperparameters:** The SlowFast sampling method introduces a non-trivial number of new hyperparameters (`Kmax`, `Whist`, `σ²_stable`, `τ_min_conf`, `τ_high_conf`, etc.). While the authors provide an ablation study (Figure 5 & 6), it raises questions about the method's sensitivity and the effort required to tune it for new models or tasks. The description in Section 3.3 is dense and would benefit greatly from a formal algorithm block (pseudocode) to improve clarity and reproducibility.

- **Oversimplification of the "Positional Principle":** The paper observes that confident tokens often cluster (Positional Principle), but the proposed method only acts on a single contiguous block `[scycle, ecycle]` at a time. It's plausible that a model might become confident about multiple, non-contiguous regions simultaneously (e.g., the beginning and end of a sentence). The current method does not seem equipped to exploit this.

- **Limited Analysis of Method Dynamics:** The paper presents strong final results, but provides little insight into the internal dynamics of the SlowFast sampler. For instance, what is the typical length of an accelerated "fast" span? How often does the fallback top-k refinement (in the fast phase) get triggered? Such analysis would provide a deeper understanding of *why* the method works so well.

**Questions:**

1.  Could you please comment on the process for selecting the hyperparameters? Were they found via a systematic search, and how sensitive is the performance to their joint configuration? How well would you expect the chosen values to transfer to a completely different dLLM architecture or a different data domain?
2.  To improve clarity and reproducibility, would it be possible to provide a pseudocode algorithm for the main SlowFast Sampling loop in the appendix? This would greatly help in understanding the interplay between the slow and fast phases and the various conditions.
3.  Regarding the Positional Principle: did you consider or experiment with strategies that could identify and decode multiple non-contiguous stable blocks in parallel during the "fast" phase? Do you believe this could offer further speedups?
4.  Can you provide any statistics from your experiments on the dynamic behavior of the sampler? Specifically, what was the average number of cycles (slow-fast pairs) per generation, and what was the average length of the `[scycle, ecycle]` spans?

---

> ### Author Response · Authors · 2025-11-21
> **Author Response to Reviewer g1Di (Weakness 1, Question 1, 2):**
>
> **W1: Methodological Complexity and Hyperparameters: The SlowFast sampling method introduces a non-trivial number of new hyperparameters ($K_{max}$, $W_{hist}$, $\sigma^2_{stable}$, $\tau_{minconf}$, $\tau_{highconf}$, etc.). While the authors provide an ablation study (Figure 5 & 6), it raises questions about the method's sensitivity and the effort required to tune it for new models or tasks. The description in Section 3.3 is dense and would benefit greatly from a formal algorithm block (pseudocode) to improve clarity and reproducibility.**
>
> **Q1: Could you please comment on the process for selecting the hyperparameters? Were they found via a systematic search, and how sensitive is the performance to their joint configuration? How well would you expect the chosen values to transfer to a completely different dLLM architecture or a different data domain?**
>
> **Q2: To improve clarity and reproducibility, would it be possible to provide a pseudocode algorithm for the main SlowFast Sampling loop in the appendix? This would greatly help in understanding the interplay between the slow and fast phases and the various conditions.**
>
>
> **Answer to W1, Q1 and Q2**: We sincerely thank the reviewer for the insightful feedback regarding the number of hyperparameters and the potential effort required for tuning. We have addressed these concerns in the following three aspects:
>
> 1. **Regarding the role of $K_{max}$ and $W_{hist}$:** we clarify that these parameters were introduced primarily to ensure theoretical rigor and boundedness. In practical implementations, these constraints are not strictly essential. Results demonstrate that our method is highly robust to parameter variations, eliminating the need for task-specific tuning. Across 13 distinct configurations of $K_{max}$ and $W_{hist}$, performance remained remarkably stable, with accuracy fluctuations contained within a narrow margin of **0.02**. Crucially, even in the worst-case accuracy scenario, our method maintains a commanding lead in efficiency: achieving **an average TPS of 14.2** compared to **Fast DLLM's 7.45**, representing a near **2$\times$** speedup while retaining competitive performance.
>
> 2. **Additional Sensitivity Experiments (Appendix A.4, Figure 7 and 8)**: To further demonstrate robustness, we have conducted and added extensive cross-model and cross-dataset experiments. specifically:
>   - We analyzed the sensitivity of $\sigma^2_{stable}$, $\tau_{minconf}$, and $\tau_{highconf}$ on the MATH, MBPP, and HumanEval benchmarks.
>   - We also included specific analytical experiments on GSM8K using the Dream model.
>
> 3. **Stability and Generalization**: The data from these additional experiments confirms the high stability of our method. We observed that a single, unified set of hyperparameters consistently performs well across a broad spectrum of models and tasks. Specifically, our extensive evaluation spans **multiple distinct model architectures (including Llada and Dream)** and **8 diverse benchmarks**, covering categories such as **Mathematics & Science, General Tasks, and Code Generation**. It is worth noting that we extended this fixed hyperparameter configuration to the state-of-the-art **Llada-Var** [1] model; as shown in the table below, it maintained superior performance without any task-specific tuning. This indicates that our method is inherently robust and does not require extensive adaptation when applied to new scenarios. All results presented in our main paper were achieved using this default parameter set.
>
> **Table 1: Performance and Efficiency of SlowFast Across dLLM Variants (dLLM-Var)**
> | Task | Method | TPS ↑ | Speedup (×) ↑ | Score ↑ |
> | :--- | :--- | :--- | :--- | :--- |
> | MATH | dLLM-Var | 9.04 | 1.00 | 32.58 |
> | | **+ SlowFast** | 31.46 | 3.48 | 34.08 |
> | MBPP | dLLM-Var | 9.96 | 1.00 | 41.80 |
> | | **+ SlowFast** | 25.59 | 2.57 | 44.40 |
> | GSM8K | dLLM-Var | 4.69 | 1.00 | 79.45 |
> | | **+ SlowFast** | 18.75 | 4.00 | 79.08 |
>
>
> We thank the reviewer again for prompting this detailed analysis, which has verified the ease of use and robustness of our proposed method.
>
>
> Additionally, to improve clarity and reproducibility, we have included the detailed pseudocode for the main SlowFast Sampling loop in **Appendix A.7 ALGORITHM: SLOWFAST SAMPLING FOR DLLMS**. This pseudocode explicitly illustrates the interaction between the slow and fast phases and clarifies the key decision conditions used in our method. Finally, we confirm that our code has been fully open-sourced on a public repository. However, in strict adherence to the double-blind review policy, we are unable to provide the repository link in this rebuttal.
>
> **[1]Yang, Y., Wang, C., Wang, S., Wen, Z., Qi, B., Xu, H., & Zhang, L. (2025). Diffusion LLM with Native Variable Generation Lengths: Let [EOS] Lead the Way. arXiv preprint arXiv:2510.24605.**

---

> ### Author Response · Authors · 2025-11-21
> **Author Response to Reviewer g1Di (continued, Weakness 2 and Question 3):**
>
> **W2: Oversimplification of the "Positional Principle": The paper observes that confident tokens often cluster (Positional Principle), but the proposed method only acts on a single contiguous block [$s_{cycle}$, $e_{cycle}$] at a time. It's plausible that a model might become confident about multiple, non-contiguous regions simultaneously (e.g., the beginning and end of a sentence). The current method does not seem equipped to exploit this.**
>
> **Q3:Regarding the Positional Principle: did you consider or experiment with strategies that could identify and decode multiple non-contiguous stable blocks in parallel during the "fast" phase? Do you believe this could offer further speedups?**
>
> **Answer to W2 and Q3:** We thank the reviewer for this insightful observation regarding the positional principle and the potential for exploiting non-contiguous stable blocks. This is an important theoretical consideration for maximizing parallel decoding.
>
> 1. **Empirical Evidence for Contiguity:** To validate our design choice, we first analyzed existing decoding dynamics in large models. Evidence suggests that decoding behavior is highly localized, minimizing the missed opportunity for non-contiguous acceleration. Citing recent work on decoding dynamics (e.g., **d2cache [2], Figure 2(b)**), **models like LLaDA-8B-Instruct tend to decode the next masked token from positions close to the most recently decoded token, with 90% of tokens falling within a distance of 10.** This confirms that the vast majority of stable regions are indeed spatially contiguous or tightly localized.
> 2. **Coverage of Rare Cases:** Based on these statistical results, we find that the occurrence of truly distant, simultaneously confident tokens is extremely rare. We did observe a few extreme cases (see **Figure 10 in Appendix A.6 CASE STUDIES ON POSITIONAL PRINCIPLE AND NON-CONTIGUOUS DECODING**) where the model might decode a token early in a sentence and then jump to one near the end. However, even in these rare instances, the distance between the last decoded token and the subsequent stable token remains constrained, ensuring that the necessary span can still be efficiently covered by the current dynamic search window of our method.
> 3. **Adaptive Handling of Jumps:** Building on the analysis of these rare cases, we wanted to rigorously test the adaptive limits of the SlowFast framework to such non-contiguous jumps. To do this, as shown in **Figure 11 in Appendix A.6**, we manually constructed cases where an instability forced the sampler to jump far ahead to a ground truth token. Our adaptive SlowFast method successfully adjusted, immediately recognizing the high-confidence region at the tail of the sequence, thus completing the decoding of multiple unstable blocks efficiently. This demonstrates that SlowFast's dynamic adjustment mechanism accommodates such non-contiguities effectively even without explicit multi-span identification.
>
> We genuinely thank the reviewer for pointing out this significant theoretical direction. We fully recognize that strategies for identifying and managing truly simultaneous, non-contiguous decoding blocks represent a crucial and fascinating area for future optimization to unlock even greater speedups. We look forward to exploring this enhancement as a primary direction in our future work.
>
> **[2] Jiang, Y., Cai, Y., Luo, X., Fu, J., Wang, J., Liu, C., & Yang, X. (2025). d $^ 2$ Cache: Accelerating Diffusion-Based LLMs via Dual Adaptive Caching. arXiv preprint arXiv:2509.23094.**

---

> ### Author Response · Authors · 2025-11-21
> **Author Response to Reviewer g1Di (continued, Weakness 3 and Question 4):**
>
> **W3:Limited Analysis of Method Dynamics: The paper presents strong final results, but provides little insight into the internal dynamics of the SlowFast sampler. For instance, what is the typical length of an accelerated "fast" span? How often does the fallback top-k refinement (in the fast phase) get triggered? Such analysis would provide a deeper understanding of why the method works so well.**
>
> **Answer to W3:** We thank the reviewer for pointing out the need for a deeper analysis of the method's internal dynamics. We have conducted a detailed statistical analysis to address this:
> - **Statistics of Dynamics:** Empirically, the average length of an accelerated "fast" span is 16.12 tokens, with a decoding speed of **10.12 tokens/step** during these phases. The observed fallback top-k refinement frequency is 78.52%.
> - **Interpretation of Fallback Frequency:** Although the fallback frequency appears high, it does not negatively impact the overall acceleration. It is important to note that fallback actions do not introduce latency compared to the baseline; they essentially revert to standard decoding speed. Therefore, the overall efficiency gain is driven entirely by the rapid parallel decoding of multiple tokens during the "fast" phases.
> - **Decoding Cycle Analysis:** The dynamics reveal a distinct pattern: after a burst of high-confidence parallel decoding (achieving 10.12 tokens/step), the model naturally transitions into a period of "cautious decoding" (fallback). This period allows the model to stabilize before accumulating enough confidence to trigger the next rapid parallel decoding phase. Thus, the acceleration is achieved through these efficient bursts, despite the frequent intervening fallback steps.
>
> **Q4: Can you provide any statistics from your experiments on the dynamic behavior of the sampler? Specifically, what was the average number of cycles (slow-fast pairs) per generation, and what was the average length of the [$s_{cycle}$, $e_{cycle}$] spans?**
>
> **Answer to Q4**: We thank the reviewer for the interest in the sampler's dynamic behavior. Experiments on the GSM8K dataset reveal that the average number of cycles (slow-fast pairs) per generation is **15.88**, with an average span length ($[s_{cycle}, e_{cycle}]$) of **16.12** tokens.
>
> To contextualize these figures, we cross-referenced them with the decoding dynamics analysis presented in our **Response to W2**. There, we observed that token generation exhibits strong locality, with approximately 90% of next-token predictions falling within a distance of 10 positions from the previously decoded token. Crucially, our observed average span length (16.12) consistently covers this **high-probability zone**. This statistical alignment demonstrates that the method is structurally optimized: the dynamic window is sufficiently wide to capture and accelerate the vast majority of contiguous stable segments in a single pass, while effectively alternating to cautious verification when uncertainty exceeds this local range.

---

### Author Response · Authors · 2025-12-02
**Summary of Rebuttal for AC Re-Assignment**

Dear Area Chair,

Given the recent AC re-assignments, we would like to offer a concise summary of the reviewer consensus and our responses to key concerns to support your assessment of our submission.

We are grateful for the reviewers' thoughtful engagement with our work and for the consistent positive feedback, reflected in the scores **(8, 6, 4, 4)** and the recognition of our work across three key dimensions:

* **Method:** "Well-motivated approach" (`Reviewer g1Di`); "Proposes a novel and practical dynamic sampling strategy with a clear underlying rationale" (`Reviewer xDJE`); "Not an arbitrary collection of heuristics but a thoughtful response to the underlying behavior" (`Reviewer g1Di`); "Conceptual clarity... making the approach interpretable" (`Reviewer xDJE`).

* **Evaluation:**
"Experiments are thorough" (`Reviewer g1Di`); "Deliver on its primary promise of acceleration" (`Reviewer g1Di`); "Comprehensive evaluation... spanning reasoning, general knowledge, and code" (`Reviewer g1Di`); "Demonstrate consistent and substantial improvements in throughput and efficiency" (`Reviewer xDJE`).

* **Significance:** "Tackles a well-defined and highly relevant problem" (`Reviewer g1Di`); "Major practical contribution to the field of efficient dLLMs inference" (`Reviewer g1Di`); "Helps improve the accuracy-efficiency frontier" (`Reviewer REmQ`); "Shows real-world relevance" (`Reviewer xDJE`).

**High-Level Advantages & Impact**:

Beyond the specific rebuttals, we wish to emphasize the distinct advantages of **SlowFast Sampling** that distinguish it from existing solutions:

1. **State-of-the-Art Speedup:** We achieve up to **15.63×** speedup on LLaDA-8B and **34.22×** when combined with caching. Crucially, as noted by `Reviewer g1Di`, showing a throughput advantage over strong autoregressive baselines like LLaMA-3 8B is a "powerful statement".

2. **Model-Agnostic Robustness:** Unlike heuristic methods requiring per-model tuning, SlowFast uses a **single, unified set of hyperparameters** that generalizes across distinct architectures (**LLaDA, Dream, dLLM-Var**) and diverse tasks.

3. **Principled Adaptivity:** By leveraging the *Three Golden Principles*, our sampler dynamically allocates compute: accelerating confident spans while cautiously verifying uncertain ones, solving the "static behavior" limitation of previous samplers.

**Resolution of Key Concerns**

We have addressed all reviewer concerns with rigorous new experiments and revisions (highlighted in **blue** in the revised PDF):

* **Robustness of Hyperparameters** (`Reviewers g1Di, dgDC`): We conducted comprehensive sensitivity analyses (**Appendix A.4, Figures 7 & 8**). We demonstrated that our default configuration remains stable across 8 diverse benchmarks and new model variants (dLLM-Var and Dream), proving the method requires no task-specific tuning.

* **Validity of Speedup vs. Naive Baselines** (`Reviewers REmQ, xDJE`): To isolate the algorithmic contribution, we compared SlowFast against a "naive step reduction" baseline (**Table 4 in Rebuttal**). The results are decisive: naive reduction causes catastrophic accuracy drops, whereas SlowFast achieves 2.2×–3.2× speedup over the baseline while maintaining or improving accuracy.

* **Theoretical Mechanism & Reproducibility** (`Reviewers g1Di, dgDC`): We addressed the "Positional Principle" concerns by adding **Case Studies (Appendix A.6)** that visually demonstrate the model's ability to adaptively handle non-contiguous stable blocks. Furthermore, we provided full **pseudocode (Appendix A.7)** to ensure complete transparency and reproducibility.

In conclusion, the reviewers show a broadly positive view of the paper, recognizing it as a "novel and practical" solution (`Reviewer xDJE`). We believe we have fully addressed the remaining questions, and we hope this work offers fresh insights to the dLLM community and fosters further advancements in this rapidly evolving field.

Best regards,

**Authors of Submission 4374**

---

### Meta-Review · Area_Chair_EmSR · 2026-01-08

**Summary:**

This paper proposes SlowFast Sampling, a training-free, dynamic decoding strategy to substantially reduce inference latency for diffusion-based large language models (dLLMs). The method is motivated by three empirically observed principles and alternates between a cautious slow phase that identifies stable regions and an aggressive fast phase that decodes these regions in parallel. Extensive experiments on two state-of-the-art dLLMs across a broad set of benchmarks demonstrate large speedups with minimal degradation in accuracy.

The reviewers agree that the paper tackles a highly relevant and timely problem—making parallel decoding for dLLMs practically efficient—and that the proposed method is empirically strong, often improving the accuracy–efficiency frontier of diffusion LMs. While some reviewers initially expressed concerns about methodological complexity, hyperparameter sensitivity, attribution of gains, and clarity of presentation, the author responses are thorough, addressing most substantive critiques with additional experiments, ablations, and clarifications.

Detailed concerns:
1. Hyperparameter complexity and sensitivity: Initially a major concern across multiple reviewers. The authors respond with extensive sensitivity analyses, cross-model/dataset experiments, and evidence that a single default parameter set generalizes well, with minimal accuracy variance.
2. Attribution of speedups: The authors clarify that SlowFast provides substantial standalone gains and include comparisons to naive step reduction, showing SlowFast achieves higher speedups and preserves accuracy.
3. “Bundle of heuristics” critique: Additional component-wise ablations (slow-only, fast-only, cache removal, variance removal) demonstrate that the gains arise from the integrated design, not isolated tricks.
4. Positional principle oversimplification: The authors provide empirical evidence that confident regions are overwhelmingly contiguous in practice and show that the method adapts even in rare non-contiguous cases.
5. Lack of internal dynamics analysis: Newly added statistics (e.g., average fast-span length ≈16 tokens, ~16 slow–fast cycles per generation, fast-phase decoding at ~10 tokens/step) directly address this issue.
6. Presentation issues: The rebuttal commits to clearer labeling, dataset-specific claims, and added speed–accuracy curves.

Reviewer scores range from marginal reject to accept. Two reviewers lean slightly negative. The most positive reviews emphasize novelty, practicality, and strong empirical validation; the more critical reviews focus on clarity, attribution, and analysis rather than fundamental flaws.

After considering all reviews and the detailed author responses, this paper represents a good contribution to efficient inference for diffusion LLMs. Although there are several key concerns raised during initial review, most of them have been largely mitigated through added analyses, ablations, and clarifications, and the resulting work provides both practical impact and conceptual insight into dynamic decoding for dLLMs. This is a borderline paper, and I am slightly leaning towards acceptance if there is space.

**Reviewer Concerns:**

see above

**Reviewer Scores:**

see above

---

### Decision · Program_Chairs · 2026-01-26

Accept (Poster)